# LUMINARK: TRAINING-FREE, RELIABLE WATERMARKING FOR GENERAL VISION GENERATIVE MODELS

## ABSTRACT

Watermarking is a fundamental technique for protecting digital visual content. However, developing a general and reliable watermarking method for vision generative models remains an open challenge due to the diversity of generative paradigms and design choices. In this paper, we introduce *Luminark*, a training-free, robust and general watermarking method for vision generative models. Our approach is built upon a novel watermark definition that leverages patch-level luminance statistics. Specifically, the service provider predefines a binary pattern together with corresponding patch-level thresholds. To detect a watermark in a given image, we evaluate whether the luminance of each patch surpasses its threshold and then verify whether the resulting binary pattern aligns with the target one. A simple statistical analysis demonstrates that the false positive rate of the proposed method can be effectively controlled, thereby ensuring reliable detection. To enable seamless watermark injection across different paradigms, we leverage the widely adopted guidance technique as a plug-and-play mechanism and develop the *watermark guidance*. This design enables Luminark to achieve generality across state-of-the-art generative models without compromising image quality. Empirically, we evaluate our approach on nine models spanning diffusion (EDM2 family), autoregressive (VAR family), and hybrid (MAR family) frameworks. Across all evaluations, Luminark consistently demonstrates high detection accuracy, strong robustness against common image transformations, and good performance on visual quality.

## 1 INTRODUCTION

Watermarking has long served as a key technique for protecting digital content in computer vision. With the rise of AI-generated media, its importance has increased substantially, driven by the increasing risks of misuse and unauthorized redistribution. However, designing a general-purpose watermarking method for vision generative models remains a significant challenge. The main difficulty arises from the diversity of modern generative modeling paradigms in computer vision——ranging from diffusion-based frameworks (Ho et al., 2020; Song et al., 2020a; Lipman et al., 2022; Song et al., 2023) to autoregressive(AR) models (Tian et al., 2024; Li et al., 2024)——each of which follows distinct design principles, neural network architectures, training objectives, and inference procedures.

Existing watermarking approaches typically either (a). develop paradigm-specific watermarking techniques (Wen et al., 2023) (e.g., methods tailored to diffusion models) that cannot be transferred to other generative frameworks, (b). apply paradigm-agnostic post-hoc watermarking strategies, such as those adopted in Stable Diffusion (Cox et al., 1997), which insert predefined patterns only after the generation process has completed, but these methods are known to be vulnerable against diverse image transformations, or (c). fine-tune the model weights on watermarked images (Zhao et al., 2023; Fernandez et al., 2023; Min et al., 2024), whose effectiveness heavily depends on the finetuning's generalization ability. Given the current research landscape——where generative paradigms have not yet converged and new frameworks continue to emerge——we argue that it is crucial to develop a general watermarking algorithm that (i). enables effective, plug-and-play injection across diverse generative paradigms, (ii). support reliable detection, (iii). remains robust against a variety of image transformations, and (iv). preserves the high perceptual quality of generated visual content.

To achieve this goal, we develop *Luminark*, a training-free, robust and general-purpose watermarking algorithm that injects signatures into patch-level luminance statistics and can be seamlessly applied

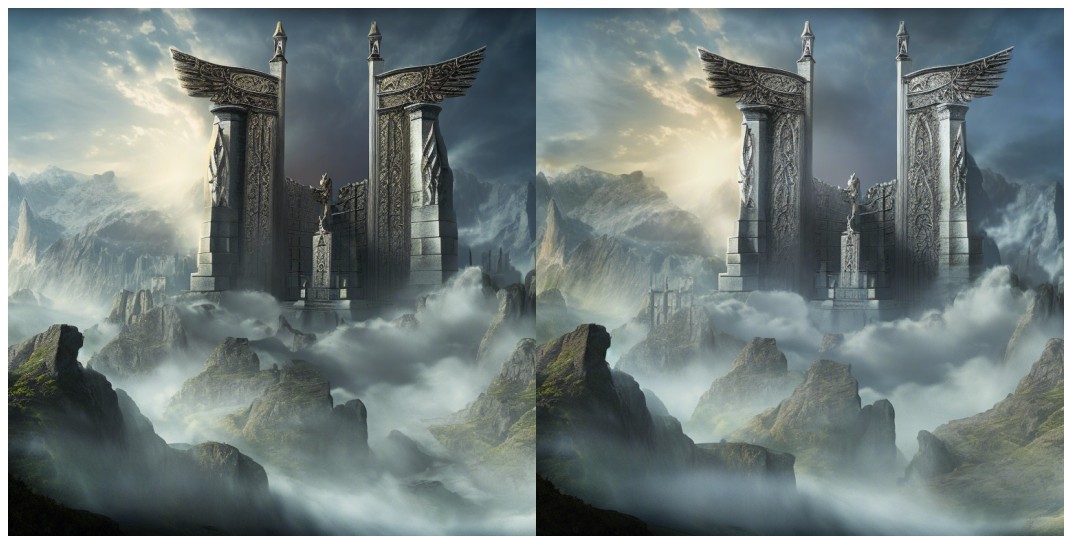

Figure 1: Luminark on Stable Diffusion[2.1]. Left: unwatermarked output. Right: watermarked output.

across all existing state-of-the-art generative paradigms. Our first contribution is the introduction of a novel watermark for image protection. Given an image, we partition it into patches and compute the luminance value of each patch. The watermark is defined as a binary pattern over these luminance values—-whether each patch's luminance exceeds a specified threshold. We mathematically show that if the binary pattern and thresholds are randomly generated, the probability that an unwatermarked image (e.g., a natural image) matches the pattern decreases exponentially with respect to the number of patches, thereby making reliable detection feasible (supporting (ii)). Moreover, since detection relies solely on patch-level statistics, it remains robust against common image transformations (supporting (iii)), such as smoothing, quantization, and compression.

From the service provider's perspective, although detection in Luminark is straightforward, injecting the predefined pattern and maintaining high-quality generation is considerably more challenging. The second contribution of our work is the design of a training-free injection algorithm that enables Luminark to operate across diverse generative paradigms and generate visually convincing outputs. The key insight is that all recent generative paradigms, despite their differences, share a common mechanism—-the guidance technique (Dhariwal & Nichol, 2021; Ho & Salimans, 2022; Karras et al., 2024a). Guidance has become a standard tool in both diffusion and autoregressive vision models, where it enhances output quality by steering the generation process toward desired outcomes. We leverage this mechanism as the entry point for watermark injection (supporting (i)) and introduce *the watermark guidance*, which directs each generation step toward alignment with the predefined pattern. This enables the watermark to be softly injected into the generation process in a plug-and-play manner, while simultaneously adjusting the content smoothly to preserve the quality of the generated image.

We conduct extensive experiments to evaluate the effectiveness of Luminark. Specifically, we test Luminark on diffusion models, autoregressive models, and hybrid solutions, including EDM2 (Karras et al., 2024b), VAR (Tian et al., 2024), and MAR (Li et al., 2024), spanning model scales from a few hundred million to several billion parameters, covering different output resolutions ($256 \times 256$ and $512 \times 512$) and diverse neural architectures (continuous and discrete tokenizers, U-Nets and Transformers). Experimental results demonstrate that Luminark largely outperforms previous baselines and preserves perceptual quality in all configurations(supporting (iv)), while simultaneously achieving high detection rates against common image transformations. These results establish Luminark as a practical and general-purpose watermarking algorithm for vision generative models.

## 2 RELATED WORKS

Existing watermarking approaches in vision can be broadly classified into three categories: Post-hoc watermarking, training-based watermarking, and watermarking tailored for a specific paradigm.

**Training-free, post-hoc watermarking** refers to early approaches that inject a watermark into an image post hoc and can be applied to any generative modeling method. To ensure imperceptibility, these methods typically transform the image into the frequency domain and adjust specific frequency coefficients to encode a signature (Bi et al., 2007; Hsieh et al., 2001; Pereira & Pun, 2000; Potdar et al., 2005). However, these approaches face a crucial trade-off between maintaining image quality and achieving robustness. Modifying low-frequency coefficients often leads to noticeable degradation in image quality, whereas modifying high-frequency coefficients makes the watermark vulnerable to local perturbations (Cox et al., 2007). Owing to its efficiency and model-agnostic nature, this approach remains widely used, including in systems such as Stable Diffusion (Rombach et al., 2021).

**Training free watermarking specialized for diffusion models** (Wen et al., 2023) serves as a pioneer work on watermarking for diffusion generative models, where a signature (a tree ring) is embedded directly into the initial noise vector that seeds the generation process. For detection, an inverse ODE solver is applied to a given image to reconstruct its initial noise, and detection is performed by verifying whether the structure of the recovered noise matches the embedded signature. Subsequent works have further refined this idea using advanced patterns (Yang et al., 2024; Ci et al., 2024; Gunn et al., 2024). However, these approaches are specifically designed for the diffusion process, which cannot be extended to other strong paradigms, such as autoregressive models.

**Training-based watermarking** injects the watermark directly into the model's weights. For example, Zhao et al. (2023); Fernandez et al. (2023); Min et al. (2024) construct a dataset of pre-watermarked images and fine-tune the neural network parameters on it, relying on the model's generalization ability to transfer the watermark to other images. This approach incurs substantial computational overhead due to fine-tuning and its effectiveness is inherently limited by the scale and quality of the pre-watermarked data. Since the finetuning operates as a black box, it is also difficult to reliably determine when the injection succeeds and when detection may fail, which is not our target setting.

## 3 LUMINARK: IMAGE WATERMARKING AS LUMINANCE CONSTRAINTS

### 3.1 WATERMARK DEFINITION

In this work, we explore a novel watermarking mechanism for general image generation based on patch-level luminance statistics. In computer vision, *luminance* refers to the perceived brightness of a pixel, derived from its RGB values (Szeliski, 2022; Gonzalez & Woods, 2018). It represents the grayscale intensity that a human observer would perceive from a colored image.

Mathematically, we first partition an image $\mathbf{x}$ of size $H \times W$ into non-overlapping patches of size $k \times k$, i.e., $\mathbf{x} = \{\mathbf{p}_1, \mathbf{p}_2, \ldots, \mathbf{p}_N\}$, where $\mathbf{p}_1$ denotes the patch in the top-left corner and $\mathbf{p}_N$ corresponds to the patch in the bottom-right corner. The total number of patches $N$ is given by $\frac{H \times W}{k^2}$. For each patch $\mathbf{p}_i$, we compute the average pixel values (normalized into $(0, 1)$ in the R, G, and B channels, denoted as $(\overline{R}_i, \overline{G}_i, \overline{B}_i)$. The luminance of the patch is then defined as:

$$l(\mathbf{p}_i) = 0.299 \cdot \overline{R}_i + 0.587 \cdot \overline{G}_i + 0.114 \cdot \overline{B}_i, \tag{1}$$

and we denote the luminance of $\mathbf{x}$ as vector $\mathbf{L}(\mathbf{x}) = (l(\mathbf{p}_1), l(\mathbf{p}_2), \ldots, l(\mathbf{p}_N))$.

To avoid confusion, we denote the generated and real images as $\mathbf{x}_{\text{gen}}$ and $\mathbf{x}_{\text{real}}$, respectively. Our goal is to inject specific patterns (i.e., signature) into the generated image $\mathbf{x}_{\text{gen}}$ such that its luminance $\mathbf{L}(\mathbf{x}_{\text{gen}})$ becomes statistically distinguishable from that of the real image $\mathbf{x}_{\text{real}}$, thereby enabling watermark detection. To construct this statistical difference, we first define a binary pattern $\mathbf{c} = (c_1, c_2, \ldots, c_N) \in \{-1, 1\}^N$ and a real-valued threshold vector $\boldsymbol{\tau} = (\tau_1, \tau_2, \ldots, \tau_N) \in (0, 1)^N$. Both values in $\mathbf{c}$ and $\boldsymbol{\tau}$ can be randomly generated using a pseudo-random number generator, fixed by the service provider and are not released to the web users.

A watermark $\mathcal{W}$ is defined by the value of $\mathbf{c}$ and $\boldsymbol{\tau}$, i.e., $\mathcal{W} = (\mathbf{c}, \boldsymbol{\tau})$. The real-valued vector $\boldsymbol{\tau}$ is used to access the "level" of luminance. For each patch $i$, we compute a decision stump of the form $\text{sgn}[l(\mathbf{p}_i) - \tau_i]$, which evaluates whether the patch's luminance exceeds $\tau_i$. The sign function $\text{sgn}(.)$ returns the sign of a real number, outputting $-1$ for negative inputs, and $+1$ for non-negative inputs. Applying this across all patches yields the binary pattern of the image $\mathbf{x}$:

$$o(\mathbf{x}) = (\text{sgn}[l(\mathbf{p}_1) - \tau_1], \cdots, \text{sgn}[l(\mathbf{p}_N) - \tau_N]) \in \{-1, 1\}^N. \tag{2}$$

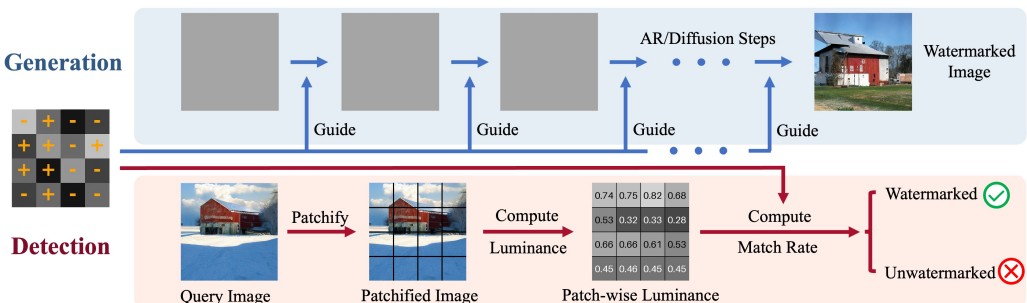

Figure 2: Illustration of Luminark. The watermark is defined as a patch-wise binary pattern (visualized by '+'/'-' symbols) over the luminance values, i.e., whether the luminance exceeds the threshold (visualized by the grayscale intensity). Generation is performed by injecting the pattern using guidance. Detection is performed by comparing the pattern in the image and the predefined pattern.

Detection is performed by comparing the similarity of the binary pattern of the image $o(\mathbf{x})$ and the predefined $\mathbf{c}$. Intuitively, since both $\mathbf{c}$ and $\boldsymbol{\tau}$ are randomly selected and known only to the provider, a natural image is unlikely to match the pattern by chance. Therefore, if the image pattern $o(\mathbf{x})$ sufficiently differs from $\mathbf{c}$, the image $\mathbf{x}$ is identified to be unwatermarked. A more rigorous detection algorithm, along with a simple statistical analysis, is presented in Section 3.2.

This approach offers several key advantages for detection. First and most notably, the watermark is designed to be imperceptible to humans, as it operates on the average luminance of each patch rather than injecting pixel-level patterns. Furthermore, the use of patch-level average luminance as the signature provides inherent robustness against several popularly studied image manipulations. The averaging calculation acts as a low-pass filter, making the signature resilient to random perturbations like added noise or artifacts from mild compression. Such modifications are unlikely to alter the average luminance of a patch enough to flip its outcome relative to the threshold, thus preserving the integrity of the watermark.

## 3.2 WATERMARK DETECTION

The objective of watermark detection is to reliably determine whether a given query image $\mathbf{x}$ has been injected with a specific predefined watermark. We begin by defining the core metric for our test. For a given image $\mathbf{x}$ and watermark $\mathcal{W} = (\mathbf{c}, \boldsymbol{\tau})$, recall that $\mathbf{x}$ is partitioned into $N$ patches $\{\mathbf{p}_i\}_{i=1}^N$. We define the **match rate**, $m(\mathbf{x}, \mathcal{W})$, as the fraction of patches whose luminance statistics align with the watermark's binary pattern $\mathbf{c}$:

$$m(\mathbf{x}, \mathcal{W}) = \frac{1}{N} \sum_{i=1}^{N} \mathbb{I}[\operatorname{sgn}(l(\mathbf{p}_i) - \tau_i) = c_i] \tag{3}$$

where $\mathbb{I}[\cdot]$ is the indicator function. We next show that, for an unwatermarked (e.g., natural) image, the match rates can be well controlled with high probability if $\mathcal{W}$ is randomly chosen.

**Proposition 1.** *Assume each $c_i$ is drawn i.i.d from* $\operatorname{Bernoulli}(\frac{1}{2})$ *with support $\{-1, 1\}$ and each $\tau_i$ is drawn i.i.d from any distribution with support $(0, 1)$, then for any fixed $\mathbf{x}$ and any $\varepsilon > 0$, we have:*

$$\operatorname{Pr}\left(m(\mathbf{x}, \mathcal{W}) > \tfrac{1}{2} + \varepsilon\right) \leq \exp\left(-2N\varepsilon^2\right),$$

Proposition 1 shows that the probability of an unwatermarked image achieving a match rate substantially exceeding 0.5 decreases exponentially with the number of patches $N$. Hence, one can set the detection threshold above 0.5, and the false positive rate, i.e., the probability of misclassifying an unwatermarked image as a watermarked one, can be controlled with statistical guarantees. The proof is provided in Appendix A.

In practice, the watermark pattern is typically fixed rather than resampled each time, and the detection threshold can be determined empirically by estimating the distribution of match rates on unwatermarked images and selecting the value that yields the desired false positive rate. Specifically, given

a large image dataset, a desired false positive rate $fpr$ (e.g., 1%), and a fixed $\mathcal{W}$, we compute the match rate for each image to obtain its empirical distribution. The threshold $T_{match}$ is then set to the $(1 - fpr)$-quantile of this empirical distribution, guaranteeing that, on average, no more than an $fpr$ fraction of natural images are misclassified. The full procedure is summarized in Algorithm 1. With a properly calibrated $T_{match}$, the final detection process for a query image $\mathbf{x}$ becomes straightforward. As described in Algorithm 2, we first compute its match rate $m(\mathbf{x}, \mathcal{W})$. If this value exceeds $T_{match}$, the image is classified as watermarked; otherwise, it is deemed unwatermarked.

---

**Algorithm 1** Setting Match Rate Threshold

---

**Input:** A set of unwatermarked images $\mathcal{D}$; A pre-defined watermark $\mathcal{W} = (\boldsymbol{c}, \boldsymbol{\tau})$; A desired false positive rate $fpr$.
**Output:** The match rate threshold $T_{match}$.
1: Let a set $M \leftarrow \emptyset$
2: **for all** image $\mathbf{x} \in \mathcal{D}$ **do**
3:     Let $\{\mathbf{p}_1, \ldots, \mathbf{p}_N\}$ be the patches of $\mathbf{x}$.
4:     $m \leftarrow \frac{1}{N} \sum_{i=1}^{N} \mathbb{I}\left[\operatorname{sgn}(l(\mathbf{p}_i) - \tau_i) = c_i\right]$
5:     Add $m$ to $M$
6: Construct the empirical cumulative density function $F(\cdot)$ $M$.
7: $T_{match} \leftarrow F^{-1}(1 - fpr)$          $\triangleright F^{-1}$ is the quantile function.
8: **return** $T_{match}$

---

**Algorithm 2** Watermark Detection

---

**Input:** An query image $\mathbf{x}$; A watermark $W = (\boldsymbol{c}, \boldsymbol{\tau})$; The match rate threshold $T_{match}$.
**Output:** A boolean value: **True** if watermarked, **False** otherwise.
1: Let $\{\mathbf{p}_1, \ldots, \mathbf{p}_N\}$ be the patches of $\mathbf{x}$.
2: $m \leftarrow \frac{1}{N} \sum_{i=1}^{N} \mathbb{I}\left[\operatorname{sgn}(l(\mathbf{p}_i) - \tau_i) = c_i\right]$
3: **return** $m > T_{match}$

---

### 3.3 WATERMARK INJECTION VIA STEP-WISE GUIDANCE

Originally proposed in the diffusion model literature, guidance (Dhariwal & Nichol, 2021; Ho & Salimans, 2022) has since become widely adopted in vision generation. For context, consider a diffusion model $D(\mathbf{x}_t, \sigma_t)$ trained to progressively denoise a noisy input $\mathbf{x}_t$ at time $t$. The sampling process is then a deterministic step-by-step denoising procedure that transforms Gaussian noise into a clean image, which can often be described by an ordinary differential equation (ODE) (Song et al., 2020b; Karras et al., 2022): $\frac{d\mathbf{x}_t}{dt} = \frac{\mathbf{x}_t - D(\mathbf{x}_t, \sigma_t)}{\sigma_t}$, $\quad \mathbf{x}_T \sim \mathcal{N}(0, \sigma_{\max}^2 \mathbf{I})$, where $\sigma_{\max}$ is the initial noise level at the start of the sampling process. In practice, this continuous process is approximated by numerical solvers such as the Euler method or Runge-Kutta methods (Süli & Mayers, 2003). Using the Euler method as an illustration, the generation process can be written as

$$\mathbf{x}_{t-1} = \mathbf{x}_t - \underbrace{\frac{\mathbf{x}_t - D(\mathbf{x}_t, \sigma_t)}{\sigma_t}}_{\text{Denoising Term}}(\sigma_t - \sigma_{t-1}), \quad t = T, \ldots, 1 \qquad (4)$$

Guidance techniques introduce additional modifications into Eqn.(4) to improve the quality of the generated images at each sampling step. For example, Classifier Guidance(CG) (Dhariwal & Nichol, 2021) modifies the update direction using an image classifier $p(y|\mathbf{x}_t)$:

$$\mathbf{x}_{t-1}^{\text{CG}} = \mathbf{x}_t^{\text{CG}} - \left[\underbrace{\frac{\mathbf{x}_t^{\text{CG}} - D(\mathbf{x}_t^{\text{CG}}, \sigma_t)}{\sigma_t}}_{\text{Denoising Term}} + \underbrace{s_t \nabla_{\mathbf{x}_t^{\text{CG}}} \log p(y|\mathbf{x}_t^{\text{CG}}; \sigma_t)}_{\text{Classifier Guidance Term}}\right](\sigma_t - \sigma_{t-1}), \quad t = T, \ldots, 1 \quad (5)$$

where $s_t$ is a scale parameter that depends on the time $t$. Shortly thereafter, researchers generalized the concept of guidance beyond classifier-based targets. Notable extensions include Classifier-Free

Guidance (Ho & Salimans, 2022) and Auto-Guidance (Karras et al., 2024a). More recently, guidance has also been directly applied to other iterative generative frameworks, including autoregressive models (Tian et al., 2024) and autoregressive-diffusion hybrid models (Li et al., 2024), yielding significant performance gains.

We believe that guidance is well-suited to our scenario for two main reasons. First, prior studies have shown that guidance can be effectively applied across diverse generative paradigms, thereby satisfying the requirement for a general-purpose watermarking mechanism. Second, its inherent training-free nature makes guidance especially suitable for watermark injection: by defining a metric that quantifies the discrepancy between an intermediate generated image and the target watermark $\mathcal{W}$, the guidance process can progressively adjust the outcome in an adaptive manner.

Inspired by these works, we develop the Watermark Guidance(WG) to softly enforce the luminance constraints $\mathcal{W}$ during generation. We first define an almost everywhere differentiable penalty function $\text{Penalty}(\mathbf{x}, \mathcal{W})$, serving as a surrogate of the match rate $m(\mathbf{x}, \mathcal{W})$ using a hinge-like loss:

$$\text{Penalty}(\mathbf{x}, \mathcal{W}) = \sum_{i=1}^{N} \max\{0, c_i \cdot (\tau_i - l(\mathbf{p}_i))\}. \tag{6}$$

The inner term $c_i \cdot (\tau_i - l(\mathbf{p}_i))$ is positive if and only if the constraint for patch $i$ is violated, and the $\max\{0, \cdot\}$ operation ensures that satisfying all constraints contributes zero loss. We replace the original guidance term in Eqn.(5) with this penalty and obtain the watermark-guided update:

$$\mathbf{x}_{t-1}^{\text{WG}} = \mathbf{x}_t^{\text{WG}} - \left[ \underbrace{\frac{\mathbf{x}_t^{\text{WG}} - D(\mathbf{x}_t^{\text{WG}}, \sigma_t)}{\sigma_t}}_{\text{Denoising Term}} + \underbrace{s_t \nabla_{\mathbf{x}_t^{\text{WG}}} \text{Penalty}(\mathbf{x}_t^{\text{WG}}, \mathcal{W})}_{\text{Watermark Guidance Term}} \right] (\sigma_t - \sigma_{t-1}), \quad t = T, \dots, 1 \tag{7}$$

The full procedure is outlined in Algorithm 3. Above, we demonstrated our watermark guidance mechanism in the context of pixel-level diffusion models using the Euler method. One potential concern is the difficulty of extending our method to more complex settings. However, it is worth noting that in most recent models, guidance is typically formulated as additive terms within the step-wise update. To integrate our approach, we can simply either (a) apply watermark guidance to these models by replacing existing guidance terms, or (b) combine it with other additive guidance by simply appending the additional term. Appendix D and E provide detailed generation pipelines in representative models and the corresponding watermark injection implementations.

---

**Algorithm 3** Luminark Injection via Guided Sampling

---

1: **Input**: Denoiser model $D(\mathbf{x}, \sigma)$, diffusion steps $T$, noise schedule $\{\sigma_t\}_{t=0}^{T}$, watermark $\mathcal{W} = (\mathbf{c}, \boldsymbol{\tau})$, watermark guidance scale $s$, threshold $T_{\text{match}}$
2: **repeat**
3:     **Initialize**: sample $\mathbf{x}_T \sim \mathcal{N}(0, \sigma_T^2 \mathbf{I})$
4:     **for** $t = T, \dots, 1$ **do**

$$\mathbf{x}_{t-1} \leftarrow \mathbf{x}_t - \left[ \underbrace{\frac{\mathbf{x}_t - D(\mathbf{x}_t, \sigma_t)}{\sigma_t}}_{\text{Denoising Term}} + \underbrace{s \nabla_{\mathbf{x}_t} \text{Penalty}(\mathbf{x}_t, \mathcal{W})}_{\text{Watermark Guidance Term}} \right] (\sigma_t - \sigma_{t-1})$$

6:     $m \leftarrow \frac{1}{N} \sum_{i=1}^{N} \mathbb{I}[\text{sgn}(l(\mathbf{p}_i) - \tau_i) = c_i]$     ▷ Compute Match Rate
7: **until** [1]$m \geq T_{\text{match}}$
8: **return** $\mathbf{x}_0$

---

## 4 EXPERIMENT

For a comprehensive comparison, we conduct experiments on nine state-of-the-art ImageNet-pretrained generative models, as summarized in Table 2. These models span diverse architectures,

---

[1]To achieve successful injection, we repeatedly generate images until the match rate exceeds the threshold. In practice, the number of repetitions is typically small, e.g., about 2.3 in the EDM2 experiment.

parameter scales, and configurations. Building on this foundation, we adopt ImageNet as our primary experimental platform. Due to space limitations, the detailed baseline descriptions, hyperparameters, visualization, and extended results on the text-to-image scenario are provided in the appendix.

## 4.1 ROBUSTNESS

Robustness aims to assess whether a watermark remains detectable after image transformations introduced by potential adversaries. In this subsection, we examine the robustness of our method against a range of practical image manipulations and compare its performance with baseline approaches.

We compare Luminark against five training-free methods: post-hoc watermarking DwtDct (Cox et al., 1997), DwtDctSvd (Cox et al., 1997), and RivaGAN (Zhang et al., 2019); recent diffusion-specific watermarking Gaussian-Shading (GS) (Yang et al., 2024) and PRC-W (Gunn et al., 2024). The post-hoc baselines are evaluated across all models, whereas the diffusion-specific baselines are evaluated only on EDMs. For all baselines, we use the official implementation if provided [2]. For our approach, the hyperparameters (e.g., patch size and watermark guidance scale) is provided in Appendix G. For each experiment, we first draw a unique watermark for each method and keep it fixed. Then, for every method, we sample 1,000 unwatermarked and 1,000 watermarked images. To assess robustness, each image is further subjected to nine different transformations: Scaling, Cropping, JPEG Compression, Filtering, Smoothing, Color Jitter, Color Quantization, Gaussian Noise, and Sharpening. The implementation for each transformation is detailed in Appendix B. We then re-apply watermark detection for each method on the transformed images and measure the detection accuracy. Each experiment is repeated 20 times with different random seeds to test the influence of watermark patterns and the sampled images. The average performance is reported.

**Experimental Results.** The experimental results are reported in Table 1. It can be seen that in all settings, our method consistently achieves accuracies of at least $\geq 95\%$, highlighting its wide applicability across different generative paradigms, and reliable detectability under diverse transformations. Relative to prior methods, Luminark significantly outperforms post-hoc baselines (DwtDct $\sim 58\%$, DwtDctSvd $\sim 65\%$, RivaGAN $\sim 82\%$). Even when compared with diffusion-specific approaches on EDMs, Luminark remains highly competitive: its average performance is nearly the same as PRC-W and only marginally below Gaussian-Shading. We attribute the success of Luminark to its use of a regional statistical signature that is inherently more robust—while individual pixel values may vary, the patch-wise statistical properties are largely preserved. Due to space limitations, we cannot test all possible transformations. A natural concern is that certain common operations, such as flipping, could lead to huge detection mistakes. This issue can be addressed with a straightforward strategy: for any given image, we apply the watermark detector to both the original and its flipped version, and then use the logical "OR" of the two outcomes as the final detection result.

## 4.2 FIDELITY

Following standard practice, we use the Fréchet Inception Distance (FID) (Heusel et al., 2017) to evaluate the generation quality of image generative models. We conduct experiments on the same set of models as in Section 4.1 and compare Luminark with Gaussian-Shading and PRC-W. The three post-hoc baselines are excluded from this evaluation due to their weak robustness. For each experiment, we first sample a unique watermark for each method and keep it fixed. Then, for each method, we generate 50,000 watermarked images and compute the FID scores against the clean image dataset. Each experiment is repeated 20 times, and the average performance is reported.

**Experimental Results.** As shown in Table 2, the FID scores of watermarked images with Luminark significantly surpass those of Gaussian-Shading and PRC-W, while closely approaching the non-watermarked references across all nine models. The differences between Luminark and the reference values are generally around 1.0. Notably, the FID of watermarked EDM2-XXL is only 2.13, demonstrating exceptionally high generation quality. In comparison, Gaussian-Shading and PRC-W nearly double the reference FID score, not to mention they are not applicable in VAR and MAR.

---

[2]Since Tian et al. (2024) did not release the inference code for VAR, we re-implemented it ourselves. The results reported here are based on our reproduction, which differs slightly from the original paper. We refer to MAR (Li et al., 2024) as a hybrid as it uses autoregressive generation with diffusion heads. For completeness, we describe Gaussian-Shading and PRC-W in detail in Appendix F.

Table 1: Detection accuracy under various image transformations. Luminark exhibits consistent robustness and performs competitively against strong baselines across all models and all attack types. Values exceeding 95% are highlighted in blue.

| Model | Method | Scaling | Cropping | JPEG Comp. | Filtering | Smoothing | Color Jitter | Color Quant. | Gauss. Noise | Sharpening | Average |
|---|---|---|---|---|---|---|---|---|---|---|---|
| EDM2-XS | DwtDct | 49.90 | 53.55 | 49.50 | 49.50 | 49.50 | 96.40 | 82.60 | 49.50 | 50.75 | 58.47 |
| | DwtDctSvd | 63.90 | 95.30 | 55.40 | 57.80 | 59.25 | 86.90 | 63.35 | 50.00 | 56.15 | 65.34 |
| | RivaGAN | 77.70 | 98.60 | 75.40 | 70.45 | 85.45 | 99.35 | 86.30 | 72.10 | 74.30 | 82.19 |
| | GS. | 99.51 | 99.50 | 99.52 | 99.50 | 99.50 | 99.49 | 99.48 | 99.50 | 99.49 | 99.50 |
| | PRC-W. | 94.00 | 99.40 | 98.35 | 93.05 | 97.20 | 99.50 | 98.40 | 91.75 | 99.40 | 96.79 |
| | Ours | 94.75 | 95.35 | 99.00 | 96.45 | 98.15 | 95.00 | 95.25 | 96.15 | 95.75 | 96.21 |
| EDM2-L | DwtDct | 49.90 | 53.60 | 49.50 | 49.50 | 49.50 | 96.60 | 82.25 | 49.50 | 50.75 | 58.51 |
| | DwtDctSvd | 63.45 | 95.60 | 55.20 | 57.60 | 59.50 | 86.00 | 63.45 | 50.00 | 56.25 | 65.23 |
| | RivaGAN | 78.10 | 99.30 | 75.25 | 70.35 | 85.35 | 99.40 | 86.05 | 72.70 | 74.30 | 82.31 |
| | GS. | 99.51 | 99.50 | 99.48 | 99.50 | 99.51 | 99.50 | 99.50 | 99.49 | 99.50 | 99.50 |
| | PRC-W. | 92.80 | 99.35 | 98.10 | 90.80 | 96.60 | 99.50 | 97.90 | 88.20 | 99.45 | 95.86 |
| | Ours | 94.65 | 95.15 | 98.60 | 96.15 | 98.00 | 94.65 | 95.15 | 96.00 | 96.00 | 95.49 |
| EDM2-XXL | DwtDct | 49.90 | 53.60 | 49.50 | 49.50 | 49.50 | 95.95 | 82.00 | 49.50 | 50.75 | 58.36 |
| | DwtDctSvd | 63.65 | 96.05 | 55.30 | 57.65 | 59.30 | 86.65 | 63.35 | 50.00 | 56.15 | 65.34 |
| | RivaGAN | 77.60 | 99.25 | 75.65 | 70.60 | 85.95 | 99.25 | 86.75 | 72.40 | 73.90 | 82.37 |
| | GS. | 99.51 | 99.50 | 99.48 | 99.50 | 99.50 | 99.49 | 99.50 | 99.50 | 99.50 | 99.50 |
| | PRC-W. | 98.00 | 99.50 | 98.60 | 95.70 | 98.90 | 99.50 | 98.35 | 89.00 | 99.35 | 97.44 |
| | Ours | 95.50 | 96.00 | 99.20 | 96.60 | 98.35 | 94.50 | 95.40 | 95.90 | 95.65 | 95.23 |
| VAR-d16 | DwtDct | 49.90 | 53.60 | 49.50 | 49.50 | 49.50 | 95.50 | 82.10 | 49.50 | 50.75 | 58.31 |
| | DwtDctSvd | 63.75 | 95.80 | 55.25 | 57.75 | 59.45 | 87.05 | 63.35 | 50.00 | 56.05 | 65.39 |
| | RivaGAN | 77.85 | 98.30 | 76.00 | 70.50 | 85.75 | 98.55 | 86.90 | 72.35 | 73.65 | 82.20 |
| | Ours | 96.00 | 95.65 | 98.85 | 96.70 | 97.85 | 95.30 | 94.50 | 96.10 | 96.40 | 95.81 |
| VAR-d30 | DwtDct | 49.90 | 53.65 | 49.50 | 49.50 | 49.50 | 96.75 | 82.15 | 49.50 | 50.75 | 58.47 |
| | DwtDctSvd | 63.50 | 95.55 | 55.30 | 57.50 | 59.30 | 86.95 | 63.55 | 50.00 | 56.10 | 65.31 |
| | RivaGAN | 77.20 | 99.40 | 75.95 | 70.90 | 86.55 | 99.30 | 86.45 | 72.50 | 73.60 | 82.43 |
| | Ours | 96.00 | 96.15 | 98.90 | 96.80 | 98.60 | 95.95 | 95.45 | 95.90 | 95.35 | 96.01 |
| VAR-d36 | DwtDct | 49.90 | 53.60 | 49.50 | 49.50 | 49.50 | 96.80 | 81.85 | 49.50 | 50.75 | 58.44 |
| | DwtDctSvd | 63.65 | 95.65 | 55.30 | 57.80 | 59.20 | 86.45 | 63.10 | 50.00 | 56.20 | 65.26 |
| | RivaGAN | 77.85 | 99.00 | 75.80 | 70.40 | 86.30 | 98.85 | 87.40 | 72.30 | 73.45 | 82.37 |
| | Ours | 96.30 | 96.25 | 98.90 | 96.05 | 97.90 | 94.50 | 95.75 | 96.40 | 96.50 | 95.95 |
| MAR-Base | DwtDct | 49.90 | 53.65 | 49.50 | 49.50 | 49.50 | 96.65 | 81.45 | 49.50 | 50.75 | 58.38 |
| | DwtDctSvd | 63.45 | 95.55 | 55.35 | 57.75 | 59.35 | 87.35 | 63.40 | 50.00 | 56.20 | 65.38 |
| | RivaGAN | 77.30 | 99.45 | 75.60 | 70.60 | 86.35 | 98.55 | 87.15 | 72.05 | 74.35 | 82.38 |
| | Ours | 95.90 | 96.75 | 99.50 | 97.15 | 97.55 | 94.95 | 95.00 | 95.90 | 96.30 | 96.56 |
| MAR-Large | DwtDct | 49.90 | 53.60 | 49.50 | 49.50 | 49.50 | 96.20 | 82.35 | 49.50 | 50.75 | 58.42 |
| | DwtDctSvd | 63.60 | 95.20 | 55.35 | 57.55 | 59.25 | 86.30 | 63.15 | 50.00 | 56.15 | 65.17 |
| | RivaGAN | 77.20 | 99.05 | 75.85 | 70.40 | 86.15 | 99.10 | 86.80 | 72.55 | 74.05 | 82.35 |
| | Ours | 95.85 | 95.35 | 99.50 | 97.00 | 98.70 | 94.85 | 94.75 | 95.25 | 95.85 | 95.79 |
| MAR-Huge | DwtDct | 49.90 | 53.55 | 49.50 | 49.50 | 49.50 | 96.80 | 82.65 | 49.50 | 50.75 | 58.52 |
| | DwtDctSvd | 63.40 | 95.35 | 55.30 | 57.75 | 59.30 | 86.00 | 63.45 | 50.00 | 56.20 | 65.19 |
| | RivaGAN | 77.05 | 98.35 | 75.20 | 70.85 | 85.70 | 98.95 | 86.65 | 72.65 | 74.05 | 82.16 |
| | Ours | 94.75 | 95.10 | 99.15 | 96.30 | 98.55 | 95.35 | 94.80 | 96.65 | 96.50 | 95.80 |

Table 2: Generation quality of watermarked images using different methods. Ref. denotes the FID of unwatermarked images generated by each model. Luminark clearly outperforms previous methods, achieving FID scores that closely approach the Ref. values.

| Model | Paradigm | Tokenizer | Resolution | Param. | FID↓ | | | |
|---|---|---|---|---|---|---|---|---|
| | | | | | GS. | PRC-W. | Ours | Ref. |
| EDM2-XS | Diffusion | Continuous | 512×512 | 0.25 B | 6.99 | 6.88 | 4.40 | 3.53 |
| EDM2-L | Diffusion | Continuous | 512×512 | 1.56 B | 5.08 | 5.00 | 2.72 | 2.06 |
| EDM2-XXL | Diffusion | Continuous | 512×512 | 3.05 B | 4.87 | 4.60 | 2.13 | 1.81 |
| VAR-d16 | AR. | Discrete | 256×256 | 0.31 B | Not Appliable | | 4.04 | 3.47 |
| VAR-d30 | AR. | Discrete | 256×256 | 2.01 B | Not Appliable | | 3.06 | 2.05 |
| VAR-d36 | AR. | Discrete | 512×512 | 2.35 B | Not Appliable | | 4.22 | 2.76 |
| MAR-Base | Hybrid | Continuous | 256×256 | 0.21 B | Not Appliable | | 3.88 | 2.31 |
| MAR-Large | Hybrid | Continuous | 256×256 | 0.48 B | Not Appliable | | 3.32 | 1.78 |
| MAR-Huge | Hybrid | Continuous | 256×256 | 0.94 B | Not Appliable | | 3.09 | 1.55 |

In Appendix C, we present visualizations of watermarked and unwatermarked images generated by the nine models with the same seed as well as watermarked Stable Diffusion, revealing several interesting phenomena. First, in some cases (e.g., the first image in Figure 15), our guidance induces only subtle modifications, such as minor variations in brightness or texture, to satisfy the watermark. In other cases (e.g., the third image in Figure 14), the model introduces additional objects to meet the constraint. This demonstrates that the models have sufficient capacity to adaptively and even creatively adjust the content to embed the watermark and maintain generation quality simultaneously.

## 4.3 ABLATION STUDY

Luminark makes two key contributions: a novel watermark pattern and a general injection algorithm. An important question is whether either component could be substituted by conventional methods, and how each individually contributes to the overall success. In this subsection, we present ablation studies addressing this question and confirm that both components are essential.

**Comparison I: Luminark's watermark with other injection methods.** We investigate whether watermark injection can be achieved through more straightforward approaches than guidance. To this end, we design two baseline methods: (i) post-hoc projection, which directly enforces a percentage of the luminance constraint only after the image has been fully generated; and (ii) hard step-wise projection, which forces a fixed percentage of patches to satisfy the constraint at every step of the generation process. We conduct our experiments on three representative models: EDM2-L, VAR-d30, and MAR-Large. As shown in Table 3, both post-hoc projection and hard step-wise injection significantly degrade image quality as they introduce obvious block artifacts. In contrast, our guidance method enables the model to adaptively balance the two objectives, yielding high-quality outputs.

**Comparison II: Guided injection with other watermarks.** We investigate whether guidance can be combined with other watermark patterns to achieve performance comparable to Luminark. It is immediately clear that post-hoc watermarks discussed previously are bound to fail as they are non-robust, and diffusion-specific watermarks are incompatible. Luminark employs a specific linear combination of RGB values to construct the watermark. An interesting question is whether alternative combinations would also be effective. To investigate this, we design five variants: using only the R channel, only the G channel, only the R channel, the RGB average, as well as a random linear combination. Experiments are conducted on EDM2-L, VAR-d30, and MAR-Large. As shown in Table 3, all alternatives lead to higher FID scores. This observation appears surprising, but can be partially understood from a perceptual perspective. Luminance has long been recognized as a dominant factor in image processing (Buchsbaum, 1980; Bovik, 2010; Simoncelli & Olshausen, 2001): principal component analyses consistently reveal that the first component of image data aligns closely with the perceptual notion of luminance. This intrinsic concentration of information in the luminance channel may explain why enforcing watermark constraints along this axis preserves generation quality more effectively than using arbitrary linear combinations of RGB values.

Table 3: Ablation study

| Model | Luminark | With other watermark | | | | | With other injection method | |
|---|---|---|---|---|---|---|---|---|
| | | R | G | B | Average | Random | Post-hoc | Hard Step-wise |
| EDM2-L | **2.72** | 3.01 | 3.39 | 4.30 | 3.48 | 3.03 | 16.70 | 92.05 |
| VAR-d30 | **3.06** | 3.08 | 3.26 | 3.74 | 3.32 | 3.52 | 7.68 | 33.25 |
| MAR-L | **3.32** | 3.50 | 3.49 | 4.89 | 3.41 | 4.06 | 6.88 | 85.15 |

## 5 CONCLUSIONS, LIMITATIONS, AND FUTURE DIRECTIONS

In this work, we presented Luminark, a reliable and general watermarking method for vision generative models. By introducing a novel watermark pattern based on patch-level luminance statistics, Luminark provides a reliable detection mechanism with a statistical guarantee. To enable practical and model-agnostic watermark injection, we leveraged the widely adopted guidance technique and extended it into a plug-and-play watermark guidance framework. Comprehensive experiments demonstrate the effectiveness of our approach. According to our experience, the current version of Luminark has two limitations that can be addressed with further research. (a). The first limitation lies in the computational cost of generation, including the repeated sampling and additional backpropagation steps. We believe this issue can be mitigated through the use of a more effective penalty function and careful optimization, such as employing sparse WG updates or adopting early stopping strategies for WG when sufficient alignment has been achieved. (b). Secondly, we believe there remains room for further improvement in generation quality, particularly for MAR models. Potential solutions include adopting more fine-grained watermark patterns, as well as exploring adaptive guidance strategies.

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

# Appendix

## Table of Contents

## A   PROOF OF PROPOSITION 1

*Proof.* We introduce the auxiliary random variables $Z_i = \mathbb{I}[\operatorname{sgn}(l(\mathbf{p}_i) - \tau_i) = c_i]$, $i = 1, \cdots, N$, so that $m(\mathbf{x}, \mathcal{W}) = \frac{1}{N} \sum_{i=1}^{N} Z_i$. We first show that the variables $\{Z_i\}_{i=1}^{N}$ are i.i.d and each follows a $\operatorname{Bernoulli}(\frac{1}{2})$ distribution with support $\{0, 1\}$. The independence follows directly from the fact that the variables $\{c_i\}_{i=1}^{N}$ are independently sampled. It is evident that the support of $Z_i$ is $\{0, 1\}$, then we have $\Pr(Z_i = 1) = \Pr(c_i = \operatorname{sgn}(l(\mathbf{p}_i) - \tau_i)) = \Pr(l(\mathbf{p}_i) \geq \tau_i) \Pr(c_i = 1) + \Pr(l(\mathbf{p}_i) < \tau_i) \Pr(c_i = -1) = \frac{1}{2} l(\mathbf{p}_i) + \frac{1}{2}(1 - l(\mathbf{p}_i)) = \frac{1}{2}$. Then we further have $Pr(Z_i = 0) = \frac{1}{2}$, and the theorem yields by directly applying the one-sided Hoeffding's inequality.   $\square$

## B   IMPLEMENTATION OF IMAGE TRANSFORMATIONS FOR ROBUST DETECTION

This appendix provides the Python code snippets used for the image manipulation attacks in our robustness evaluation. Each function takes a watermarked image as a NumPy array (in BGR channel order, as used by OpenCV) and returns the transformed image. The specific parameters used in our experiments are hardcoded in these functions for clarity and reproducibility.

**Scaling**

```python
def scaling_attack(image: np.ndarray) -> np.ndarray:
    """Downscales to 96x96, then scales back to original size."""
    original_size = (image.shape[1], image.shape[0])
    downscaled = cv2.resize(image, (96, 96),
     ↪  interpolation=cv2.INTER_LINEAR)
    restored = cv2.resize(downscaled, original_size,
     ↪  interpolation=cv2.INTER_LINEAR)
    return restored
```

**Cropping**

```python
def cropping_attack(image: np.ndarray) -> np.ndarray:
    """Crops a 2-pixel border and resizes back to original."""
    original_size = (image.shape[1], image.shape[0])
    # For a 512x512 image, this crops to a 510x510 region from
     ↪  (2,2)
    cropped = image[2:-2, 2:-2]
    restored = cv2.resize(cropped, original_size,
     ↪  interpolation=cv2.INTER_LINEAR)
    return restored
```

**JPEG Compression**

```python
def jpeg_compression_attack(image: np.ndarray) -> np.ndarray:
    """Applies JPEG compression with a quality factor of 50."""
    pil_image = Image.fromarray(cv2.cvtColor(image,
     ↪  cv2.COLOR_BGR2RGB))
    buffer = io.BytesIO()
    pil_image.save(buffer, format='JPEG', quality=50)
    buffer.seek(0)
    compressed_pil = Image.open(buffer)
    compressed_cv = cv2.cvtColor(np.array(compressed_pil),
     ↪  cv2.COLOR_RGB2BGR)
    return compressed_cv
```

**Filtering (Median)**

```python
def filtering_attack(image: np.ndarray) -> np.ndarray:
    """Applies a median filter with an 11x11 kernel."""
    return cv2.medianBlur(image, 11)
```

**Smoothing (Gaussian Blur)**

```python
def smoothing_attack(image: np.ndarray) -> np.ndarray:
    """Applies a Gaussian blur with a 9x9 kernel and sigma=15."""
    return cv2.GaussianBlur(image, (9, 9), 15)
```

**Color Jitter**

```python
def color_jitter_attack(image: np.ndarray) -> np.ndarray:
    """Applies random color jitter with a factor of 0.1."""
    # Brightness, Saturation, Hue jitter in HSV space
    hsv = cv2.cvtColor(image,
    ↪   cv2.COLOR_BGR2HSV).astype(np.float32)
    hsv[:, :, 0] *= (1 + random.uniform(-0.1, 0.1)) # Hue
    hsv[:, :, 1] *= (1 + random.uniform(-0.1, 0.1)) # Saturation
    hsv[:, :, 2] *= (1 + random.uniform(-0.1, 0.1)) # Brightness
    np.clip(hsv[:, :, 0], 0, 179, out=hsv[:, :, 0])
    np.clip(hsv[:, :, 1:], 0, 255, out=hsv[:, :, 1:])
    jittered = cv2.cvtColor(hsv.astype(np.uint8),
    ↪   cv2.COLOR_HSV2BGR)

    # Contrast jitter using Pillow
    pil_img = Image.fromarray(cv2.cvtColor(jittered,
    ↪   cv2.COLOR_BGR2RGB))
    enhancer = ImageEnhance.Contrast(pil_img)
    contrast_factor = 1 + random.uniform(-0.1, 0.1)
    final_image = enhancer.enhance(contrast_factor)

    return cv2.cvtColor(np.array(final_image), cv2.COLOR_RGB2BGR)
```

**Color Quantization**

```python
def color_quantization_attack(image: np.ndarray) -> np.ndarray:
    """Reduces the number of colors to 64 using k-means in CIELAB
    ↪   space."""
    lab_image = cv2.cvtColor(image, cv2.COLOR_BGR2LAB)
    pixel_data = np.float32(lab_image.reshape((-1, 3)))
    criteria = (cv2.TERM_CRITERIA_EPS +
    ↪   cv2.TERM_CRITERIA_MAX_ITER, 20, 1.0)
    _, labels, centers = cv2.kmeans(pixel_data, 64, None,
    ↪   criteria, 10, cv2.KMEANS_RANDOM_CENTERS)
    centers = np.uint8(centers)
    quantized_lab =
    ↪   centers[labels.flatten()].reshape(lab_image.shape)
    return cv2.cvtColor(quantized_lab, cv2.COLOR_LAB2BGR)
```

**Gaussian Noise**

```python
def gaussian_noise_attack(image: np.ndarray) -> np.ndarray:
    """Adds Gaussian noise with mean=0 and std=25."""
    noise = np.random.normal(0, 25, image.shape)
    noisy_image = np.clip(image.astype(np.float32) + noise, 0,
    ↪   255)
    return noisy_image.astype(np.uint8)
```

**Sharpening**

```python
def sharpening_attack(image: np.ndarray) -> np.ndarray:
    """Applies an Unsharp Mask filter with radius=5 and
    ↪   strength=300%."""
```

```
756    pil_image = Image.fromarray(cv2.cvtColor(image,
757    ↪  cv2.COLOR_BGR2RGB))
758    # 'percent' controls strength, 'threshold' is left at default
759    sharpened = pil_image.filter(ImageFilter.UnsharpMask(radius=5,
760    ↪  percent=300))
761    return cv2.cvtColor(np.array(sharpened), cv2.COLOR_RGB2BGR)
```

## C  VISUALIZATION OF THE WATERMARKED IMAGES

For all the figures in this section, $\mathcal{W} = (\mathbf{c}, \boldsymbol{\tau})$ is fixed within each experiment. Images generated by the vanilla pipeline are placed at the top, watermarked images with Luminark using the same seeds are placed in the middle, and visualization of $\mathcal{W} = (\mathbf{c}, \boldsymbol{\tau})$ are placed at the bottom. In the bottom row, each cell corresponds to a patch: grayscale intensity encodes $\tau$ (lighter = larger luminance); +/– symbols denote $c$; symbol color indicates whether the luminance constraint is satisfied (green) or violated (red) in the corresponding watermarked image. Since Stable Diffusion is a diffusion-based model, it is naturally compatible with guidance. We directly extend our approach to the text-to-image scenario.

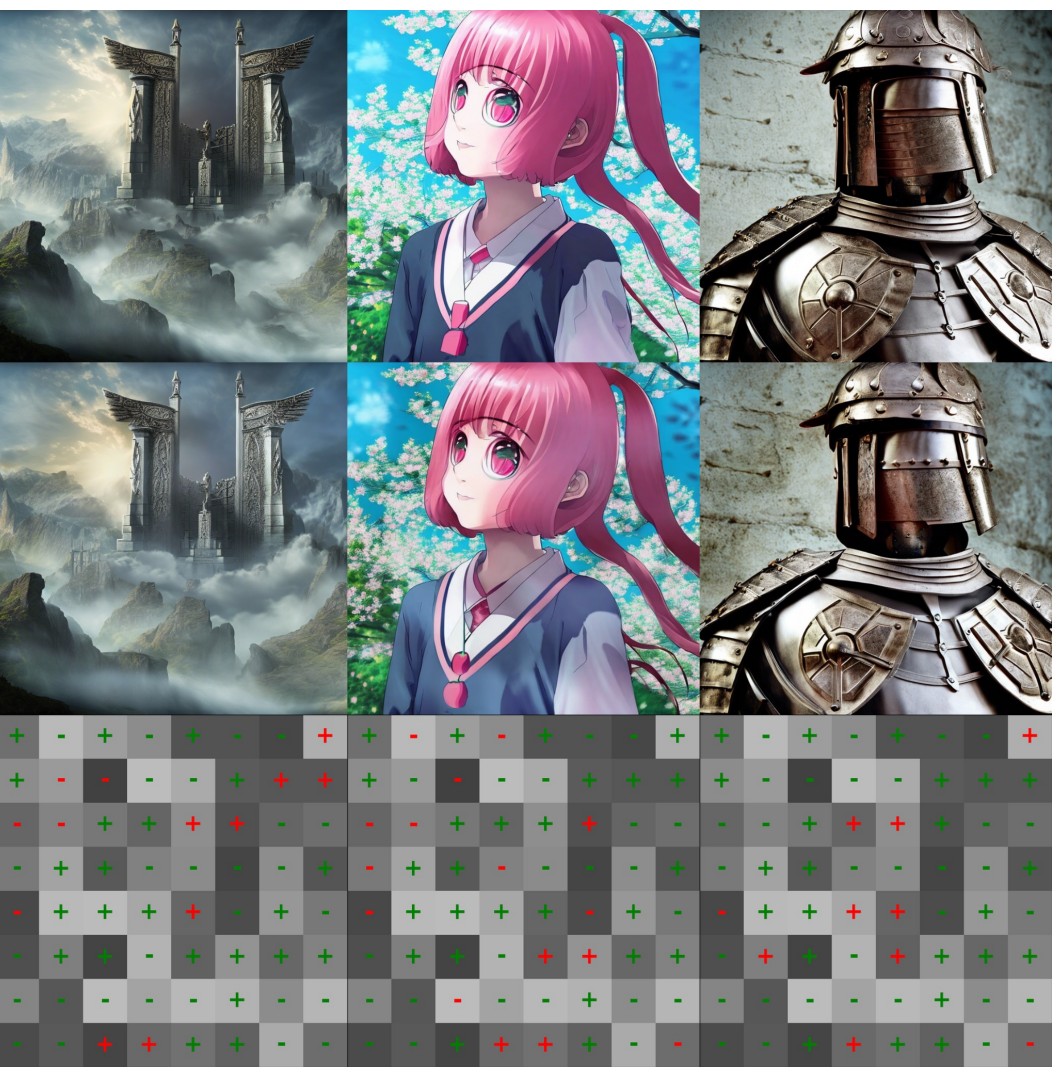

Figure 3: Visualization of Stable-Diffusion Version2.1: Prompt (left): The Gates of Valhalla, grand and imposing, with Valkyries flying around, Norse mythology, epic, divine lighting, matte painting, masterpiece. Prompt (mid): A cheerful anime girl with vibrant pink hair and large expressive eyes, wearing a school uniform, cherry blossoms falling, digital art, by Makoto Shinkai. Prompt (right): A stoic Roman centurion in full armor, standing guard, detailed armor and helmet, realistic, historical, cinematic shot.

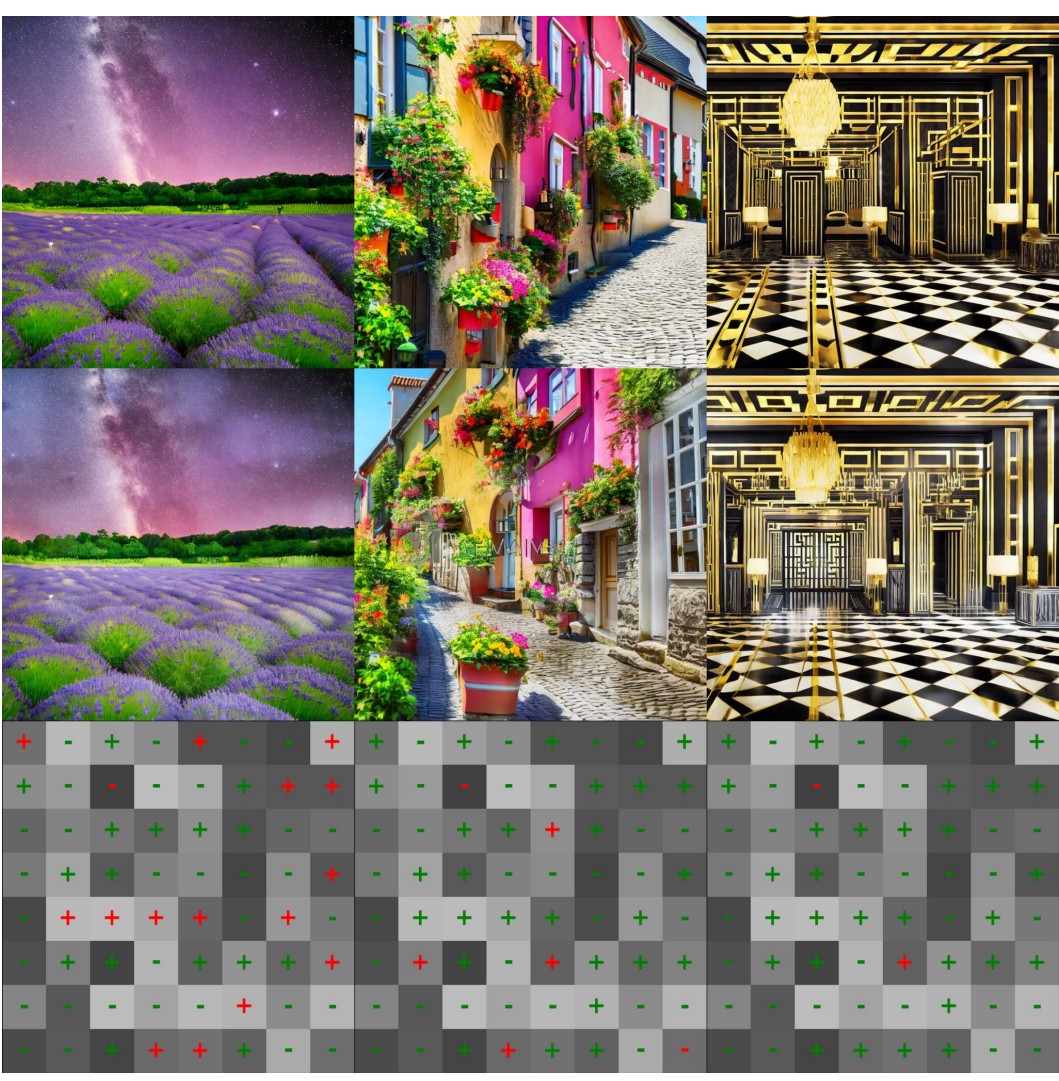

Figure 4: Visualization of Stable-Diffusion Version2.1 Prompt (left): A field of glowing lavender under the Milky Way, astrophotography, stunning, magical, long exposure. Prompt (mid): A quaint, cobblestone alleyway in a European village, colorful houses with flower boxes, charming, sunny day.Prompt (right): A luxurious Art Deco hotel lobby, geometric patterns, gold and black color scheme, elegant, 1920s style.

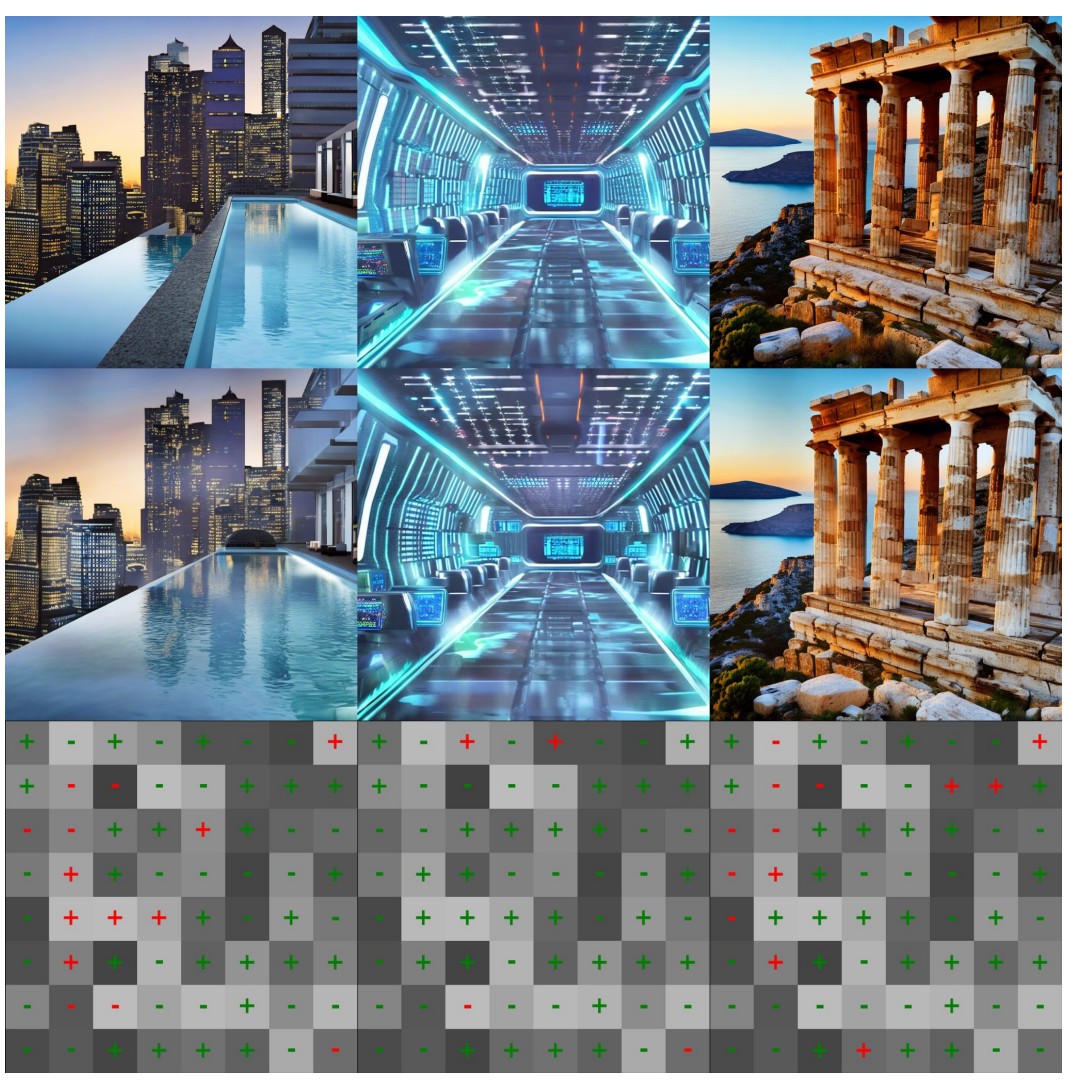

Figure 5: Visualization of Stable-Diffusion Version2.1: Prompt (left): An infinity pool on a rooftop overlooking a modern city skyline at dusk, luxurious, serene, beautiful view. Prompt (mid): The interior of a futuristic spaceship bridge, holographic displays, panoramic view of space, clean design, sci-fi. Prompt (right): An ancient, ruined Greek temple on a cliffside, overlooking the Aegean Sea, historical, majestic, golden hour.

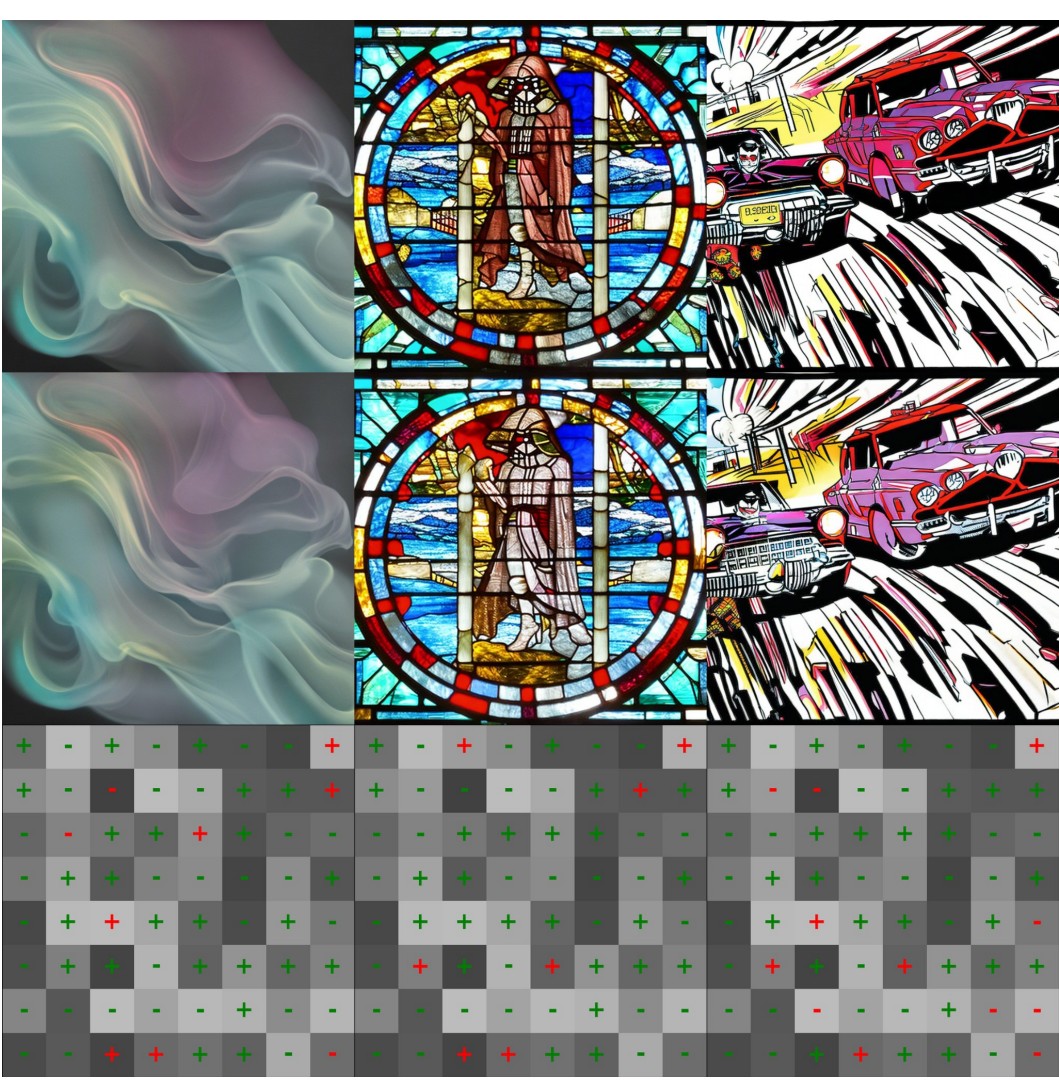

Figure 6: Visualization of Stable-Diffusion Version2.1: Prompt (left): "Serenity" visualized as flowing, pastel-colored liquid smoke, on a dark background, abstract art, calming, high resolution. Prompt (mid): A stained glass window depicting a scene from Star Wars, intricate, colorful, beautiful. Prompt (right): A car chase scene in a comic book art style, dynamic action lines, bold colors, halftone patterns, by Jack Kirby.

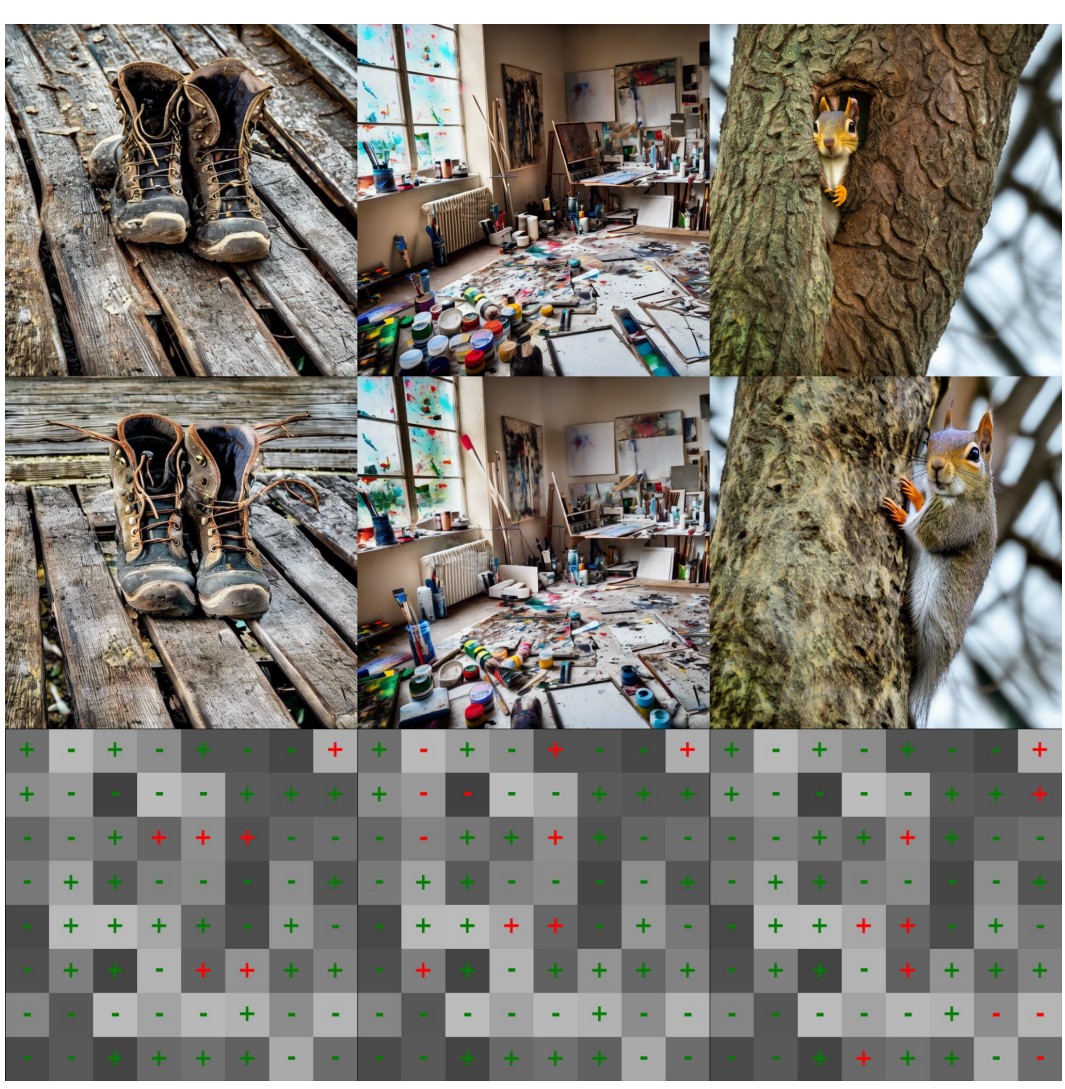

Figure 7: Visualization of Stable-Diffusion Version2.1 Prompt (left): A pair of worn-out hiking boots, covered in mud, resting on a wooden porch after a long journey, telling a story. Prompt (mid): An artist's messy studio, canvases, paint tubes, brushes scattered everywhere, creative chaos, natural light from a large window. Prompt (right): A squirrel curiously peeking from behind a tree, soft background blur (bokeh), wildlife photography.

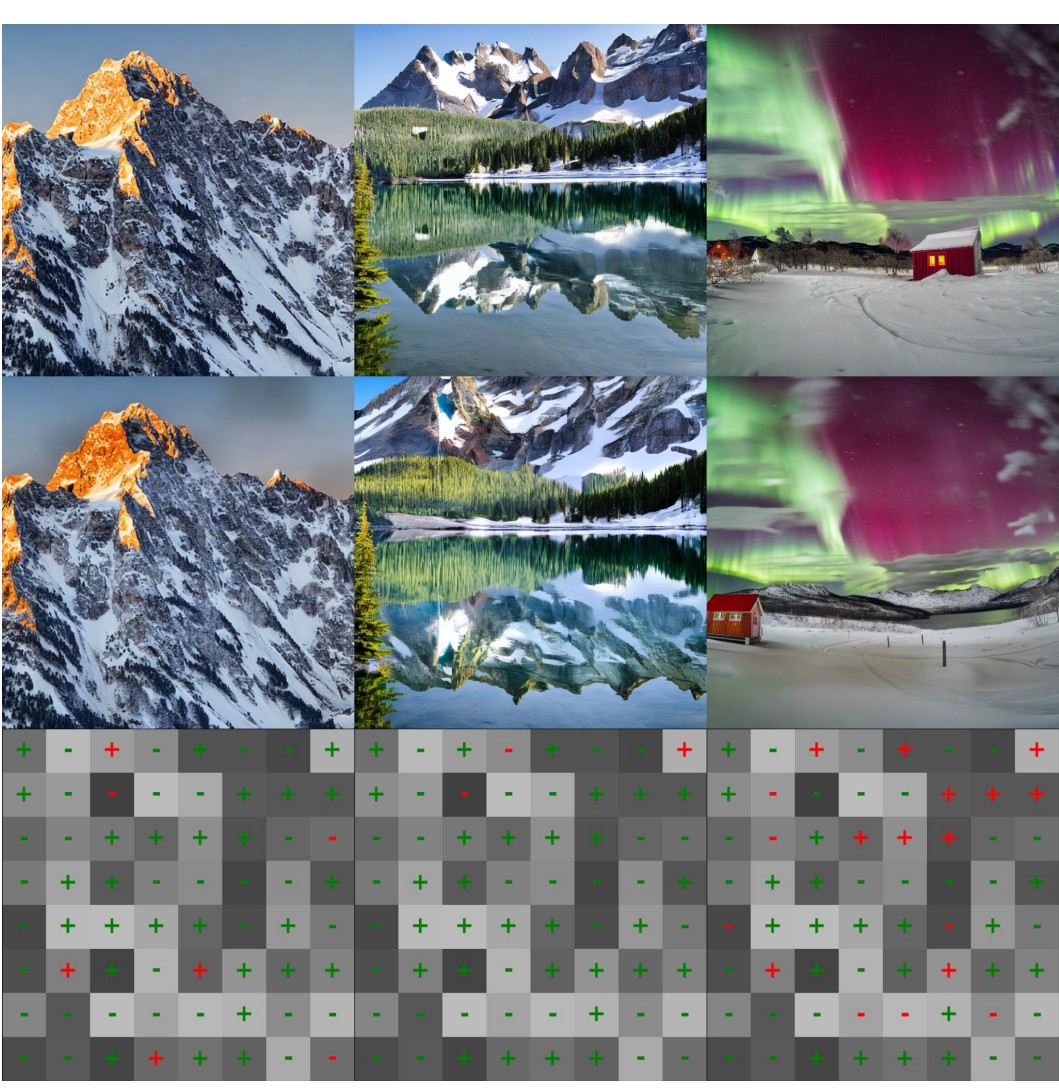

Figure 8: Visualization of Stable-Diffusion Version2.1: Prompt (left): Breathtaking panoramic photograph of the Swiss Alps at sunrise, golden light hitting the snow-capped peaks, f/16, wide-angle lens, professional landscape photography, 8k, hyper-detailed. Prompt (mid): A crystal-clear alpine lake in Canada, perfectly reflecting majestic, snow-dusted mountains, calm water, early morning light, professional landscape photo, sharp focus. Prompt (right): A vibrant aurora borealis (Northern Lights) dancing over a snow-covered landscape in Norway, with a small red cabin, professional night photography, shot with a wide-angle f/2.8 lens.

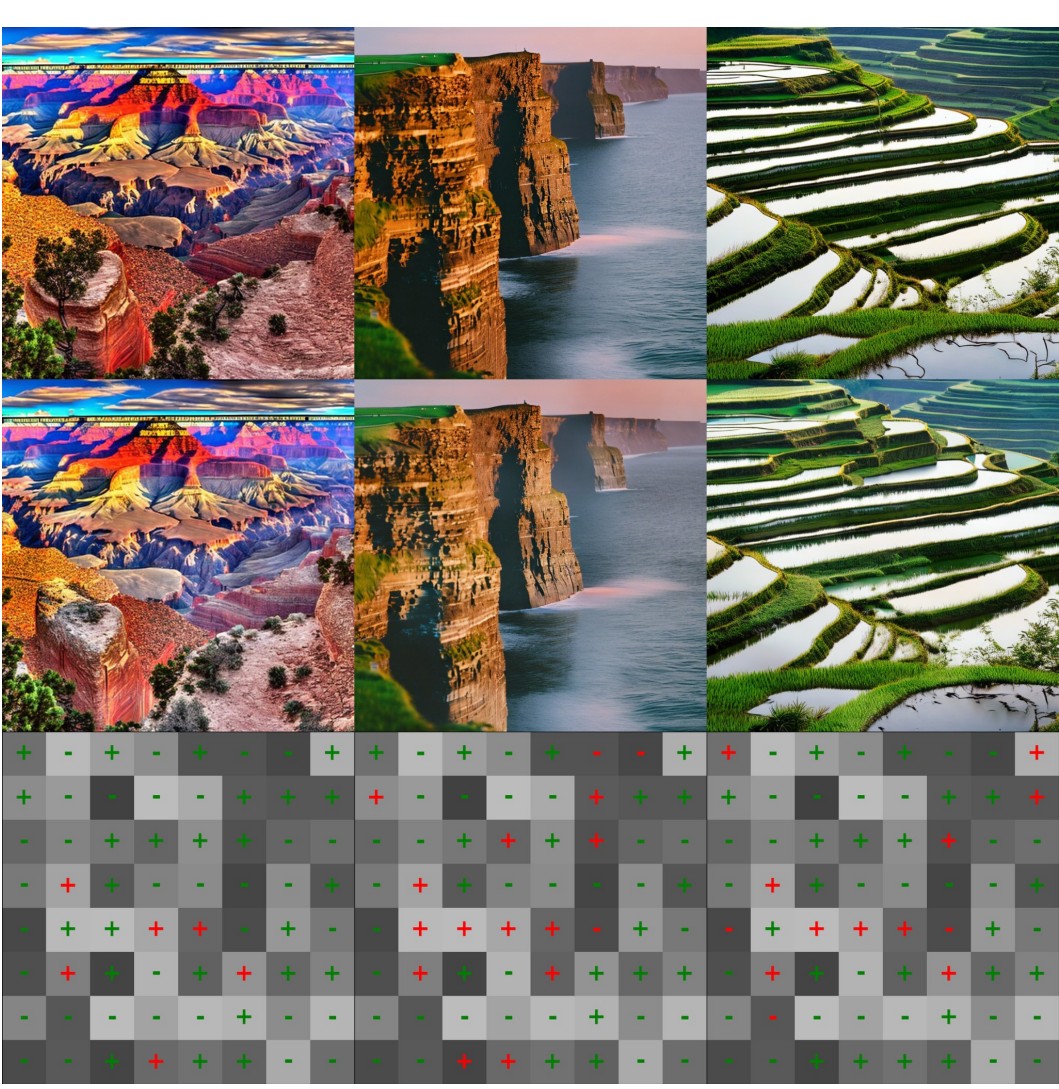

Figure 9: Visualization of Stable-Diffusion Version2.1: Prompt (left): The immense scale of the Grand Canyon at sunrise, layers of red and orange rock illuminated, epic panoramic view, high dynamic range (HDR) photo. Prompt (mid): Stunning sunset over the cliffs of Moher, Ireland, the sun dipping below the horizon casting a warm orange glow, shot on Kodak Portra 400 film, film grain, cinematic. Prompt (right): Rice terraces in Sapa, Vietnam, during the golden hour, beautiful layers and reflections in the water, lush green, professional travel photography.

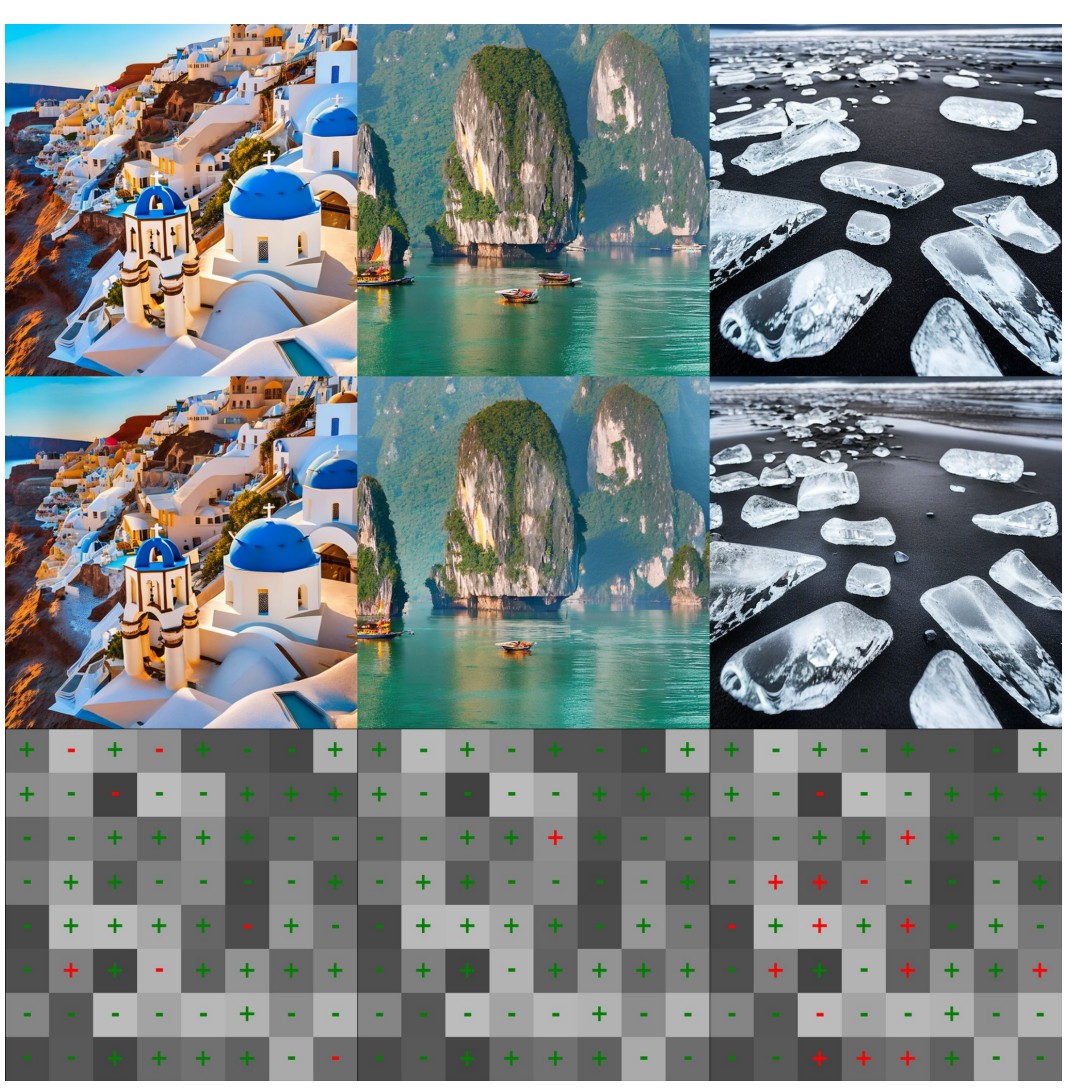

Figure 10: Visualization of Stable-Diffusion Version2.1: Prompt (left): A charming coastal town in Santorini, Greece, iconic white buildings with blue domes overlooking the calm Aegean Sea, golden hour lighting, sharp details, travel magazine photo. Prompt (mid): The unique limestone karsts of Halong Bay, Vietnam, emerging from the emerald water, traditional junk boats sailing by, misty morning, atmospheric photo. Prompt (right): A close-up, wide-angle shot of a surreal black sand beach in Iceland, with chunks of glistening ice washed ashore like diamonds, soft morning light, intricate details.

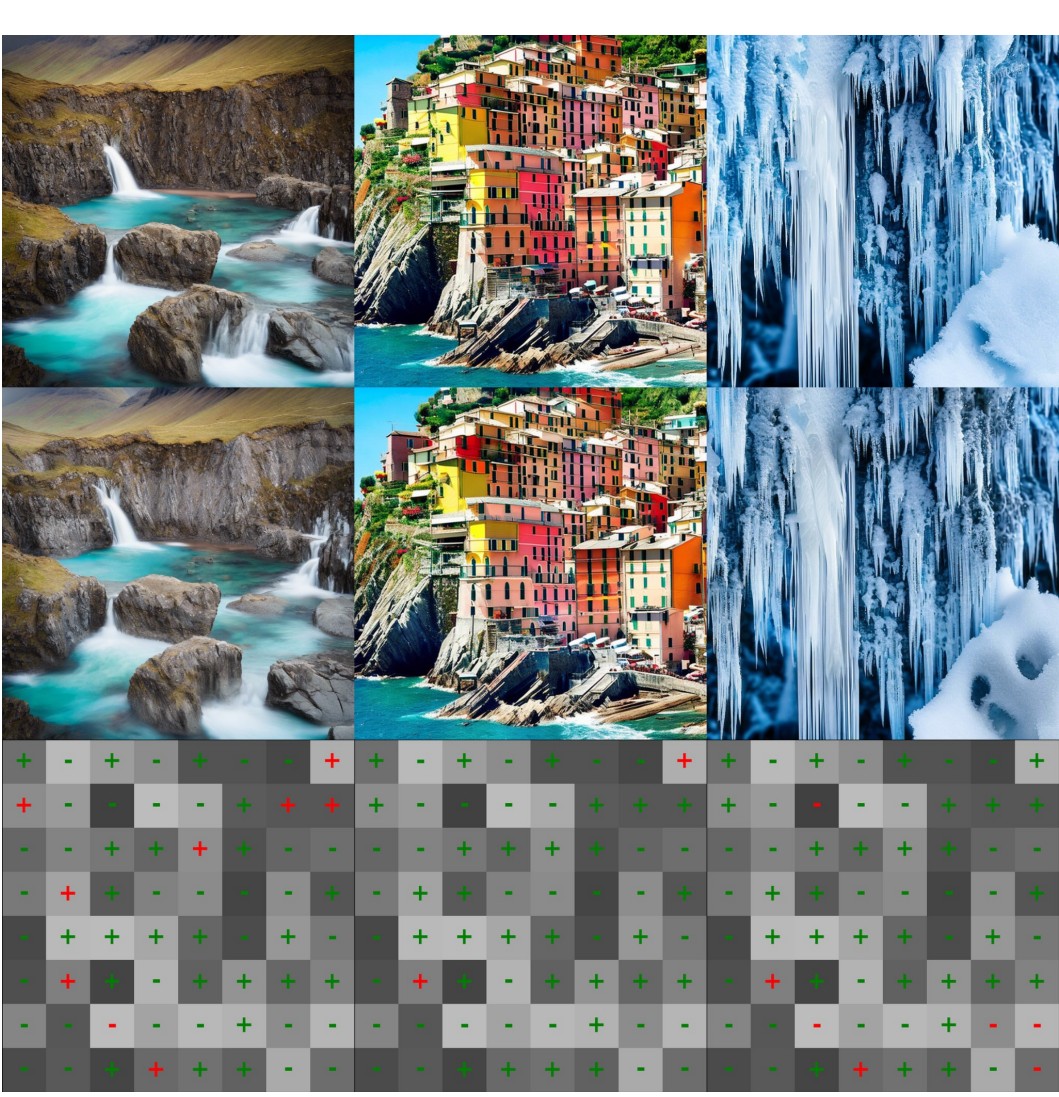

Figure 11: Visualization of Stable-Diffusion Version2.1: Prompt (left): The Fairy Pools on the Isle of Skye, Scotland, with their crystal clear blue water and rocky waterfalls, moody and magical lighting, professional photography. Prompt (mid): The vibrant, colorful houses of Cinque Terre, Italy, clinging to a cliffside above the Ligurian Sea, bright sunny day, professional travel photo. Prompt (right): A frozen waterfall in winter, with intricate icicles forming a natural sculpture, cold blue tones, macro details, sharp focus, professional nature photography.

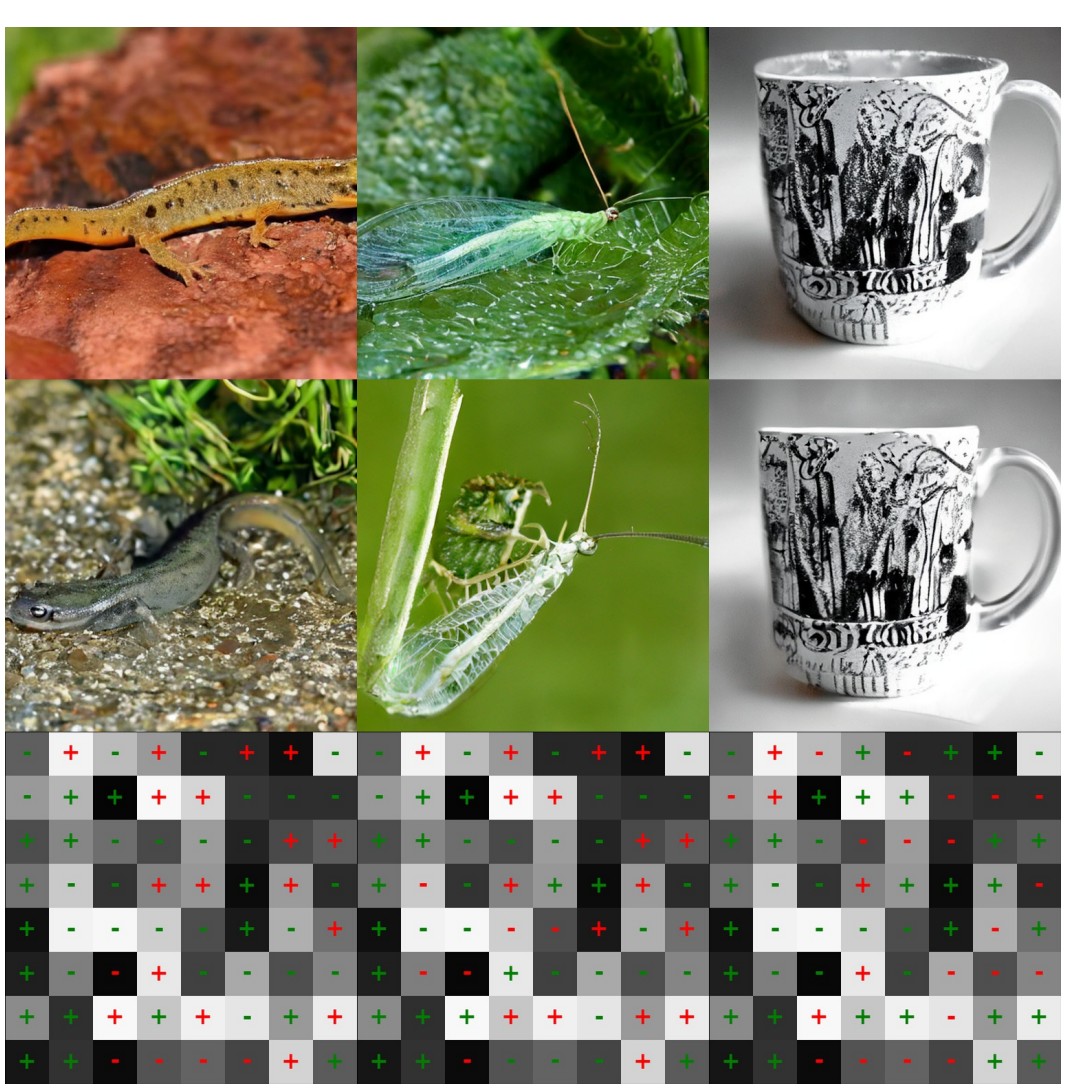

Figure 12: Visualization of EDM2-XS.

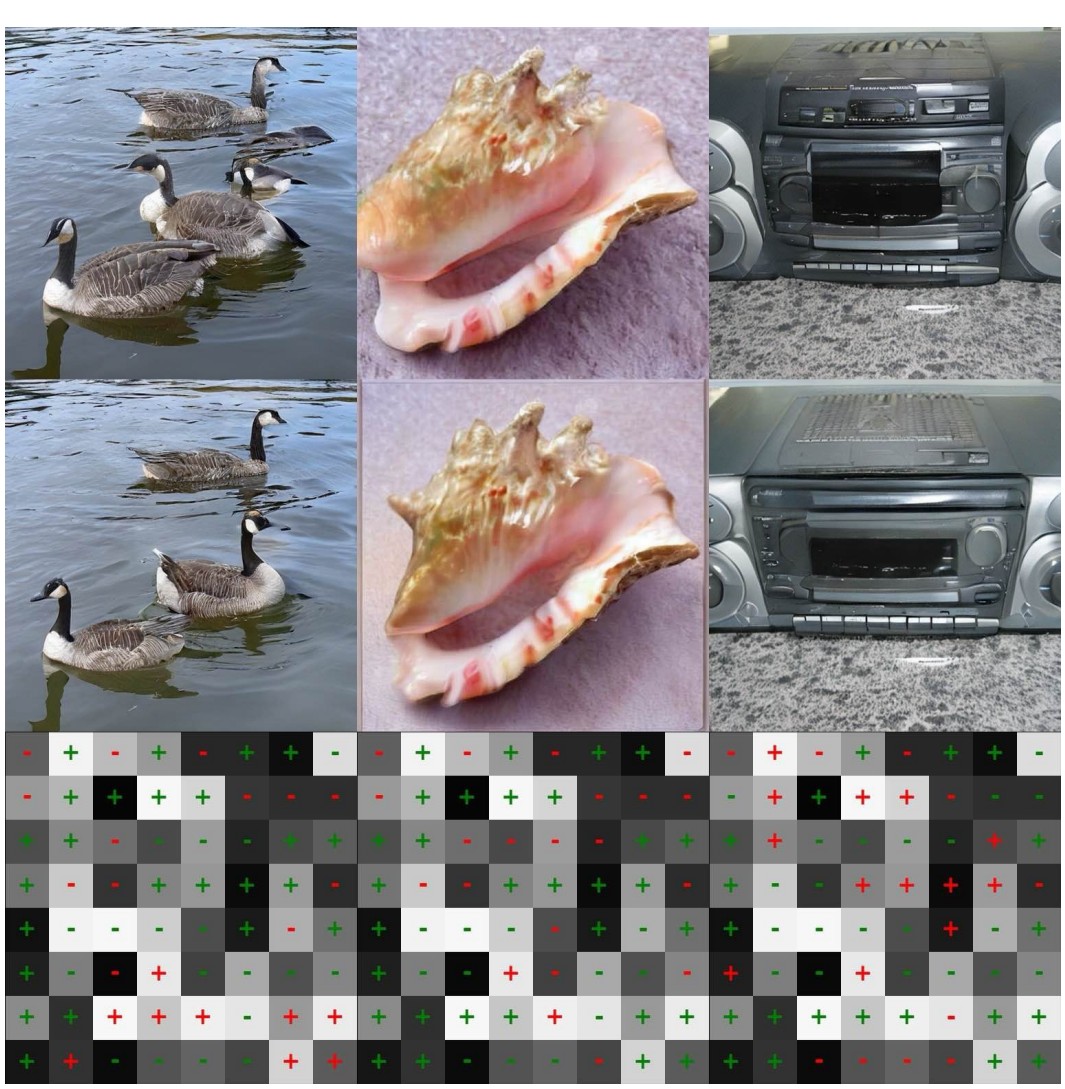

Figure 13: Visualization of EDM2-L.

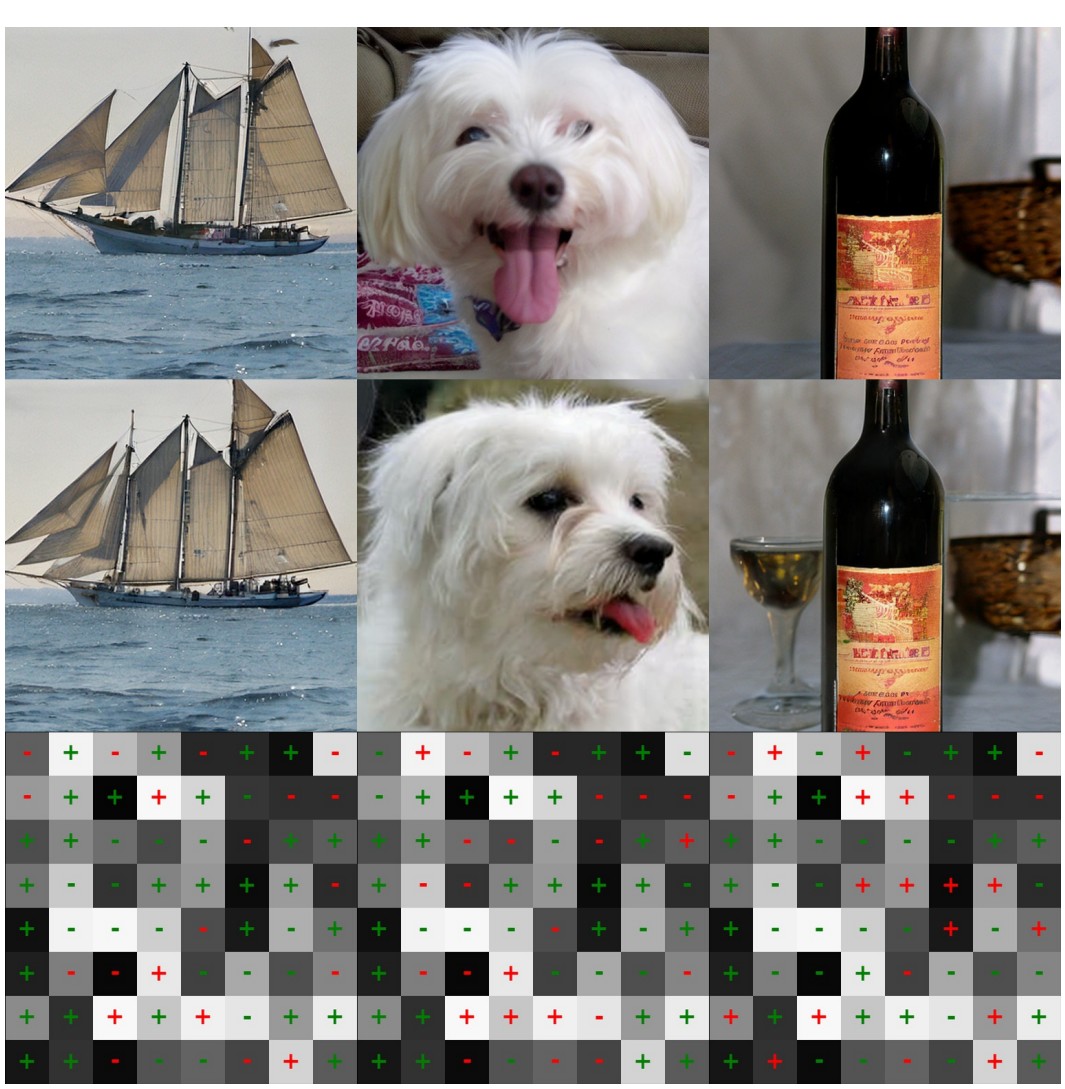

Figure 14: Visualization of EDM2-XXL.

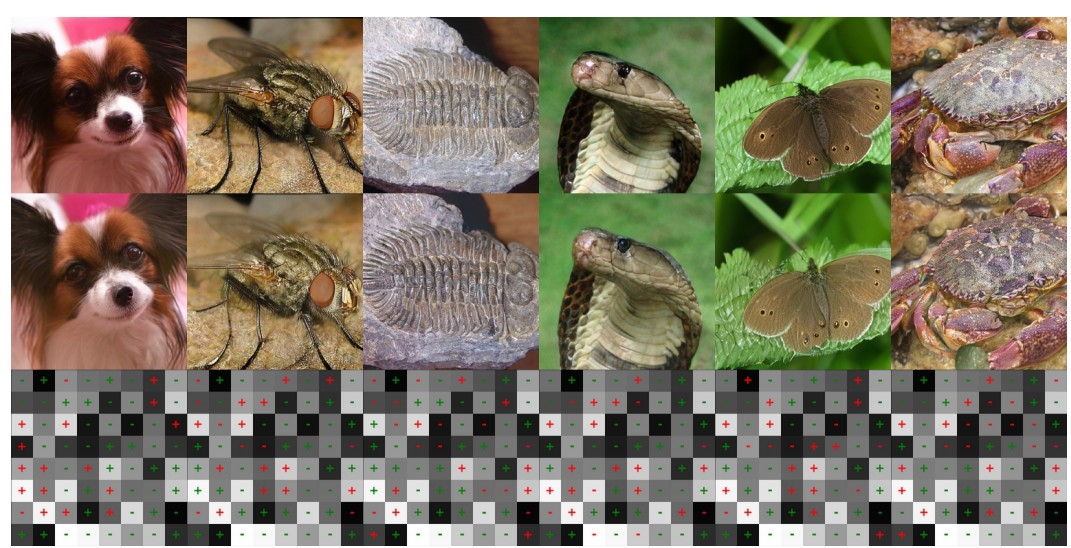

Figure 15: Visualization of VAR-d16.

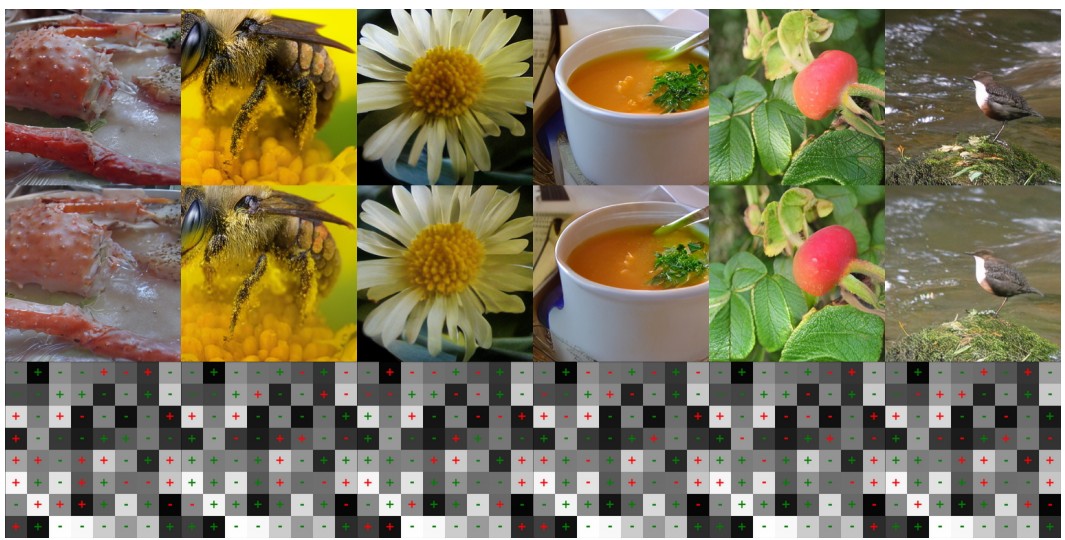

Figure 16: Visualization of VAR-d30.

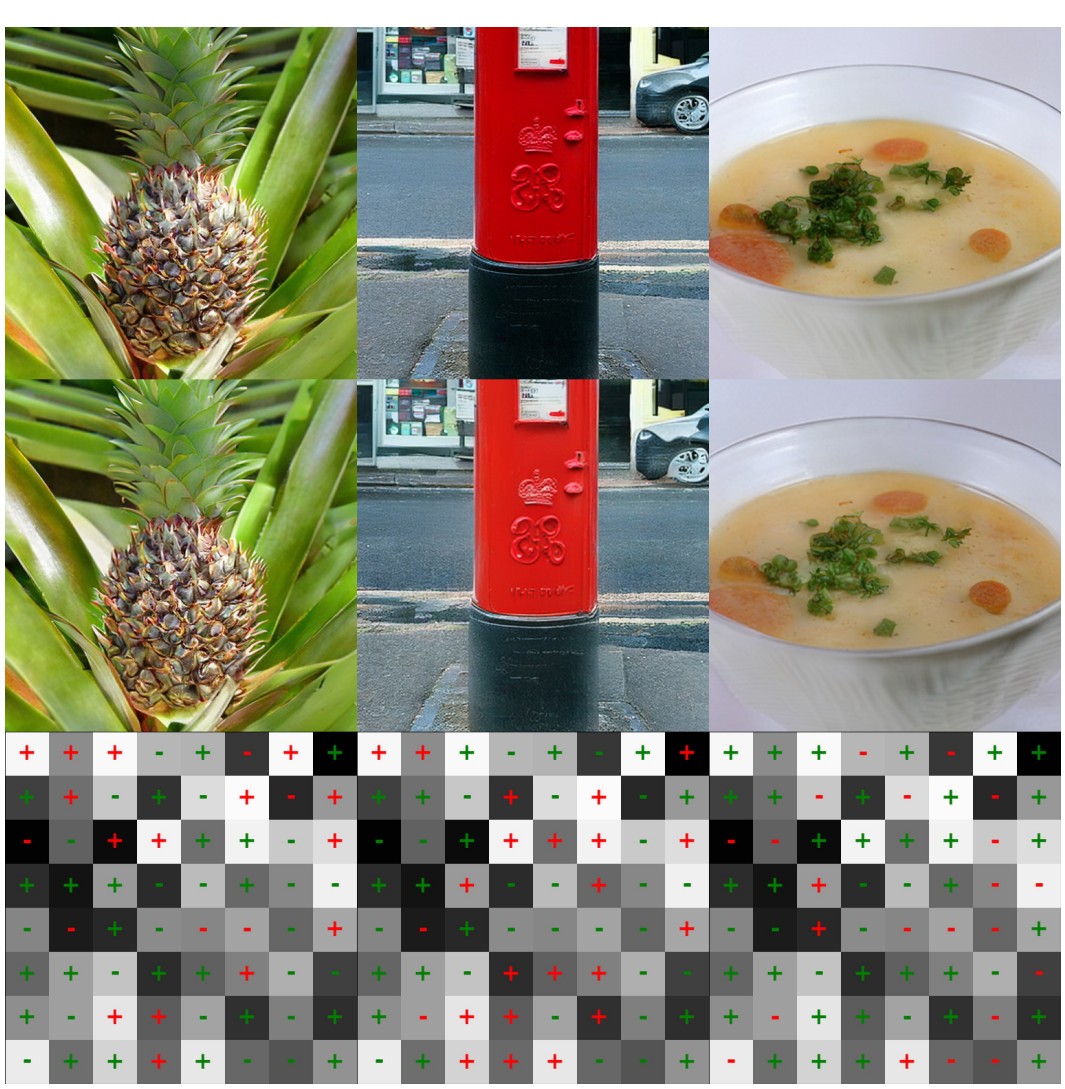

Figure 17: Visualization of VAR-d36.

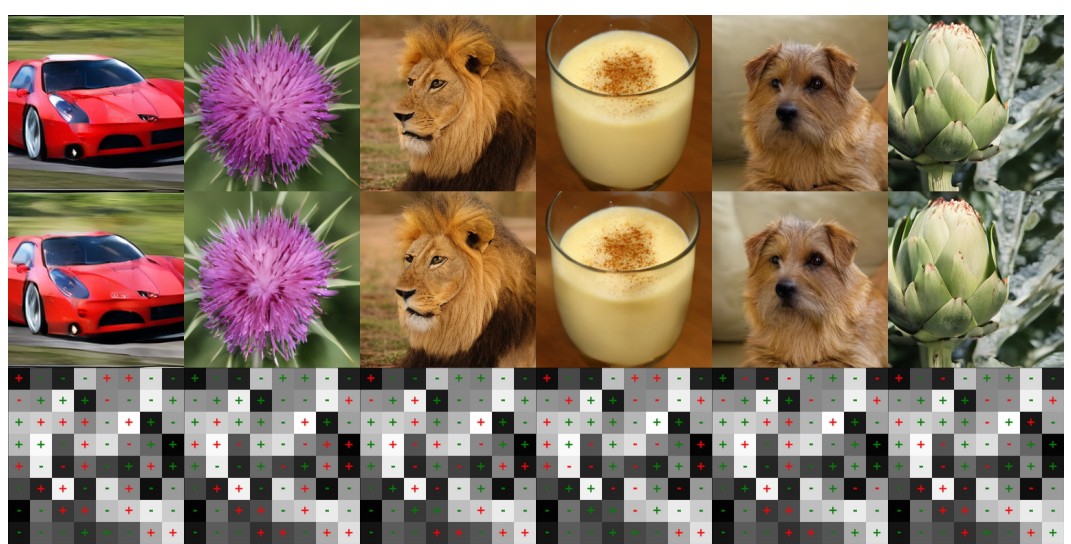

Figure 18: Visualization of MAR-B.

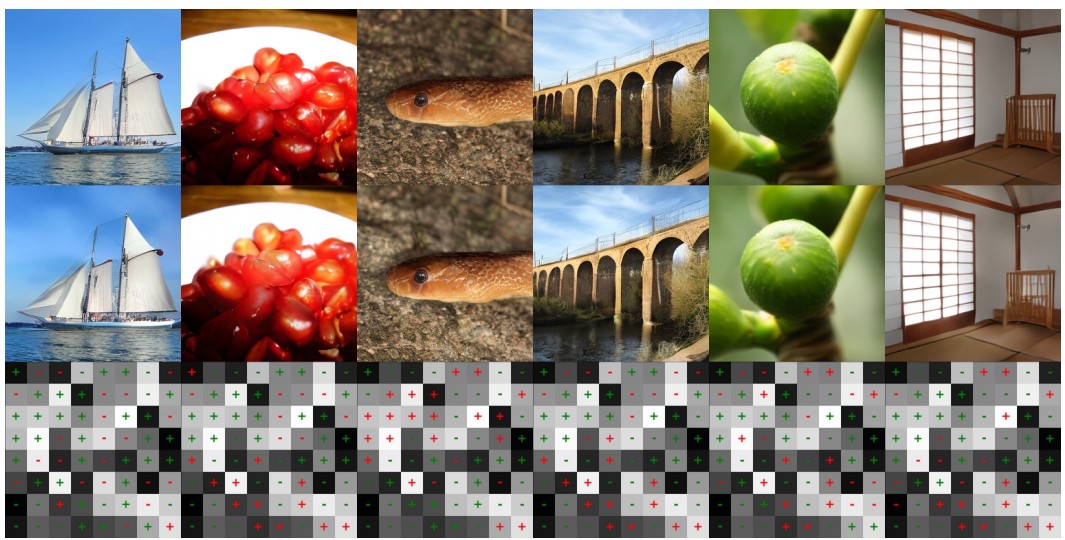

Figure 19: Visualization of MAR-L.

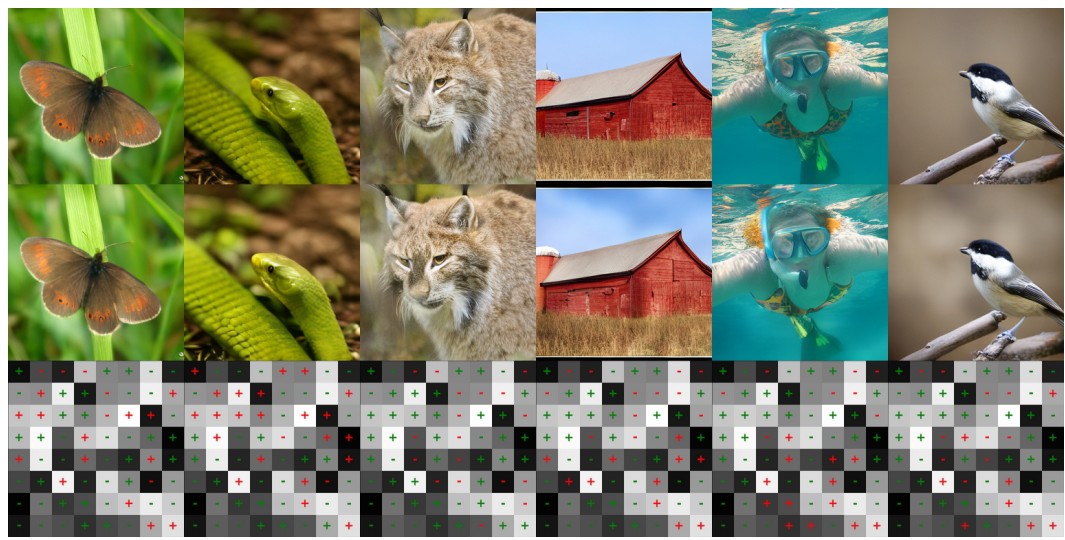

Figure 20: Visualization of MAR-H.

## D GUIDED SAMPLING FOR LATENT DIFFUSION MODELS

Algorithm 3 introduces a simple version of watermark injection for pixel-space diffusion models. While effective, state-of-the-art diffusion models typically operate in the latent space (Karras et al., 2024b). In these models, the diffusion process unfolds entirely in the latent domain, and the final image is produced by a decoder network DEC that projects the latent representation back into pixel space. It is easy to see that this framework does not affect our watermark detection algorithm. However, the generation procedure in Algorithm 3 requires adaptation. For latent diffusion models, at each sampling step $t$, we decode the latent vector $\mathbf{z}_t$ into pixel space using the decoder, enabling computation of the penalty term. Since the mapping from a latent vector to its corresponding penalty value is almost everywhere differentiable, we can still compute $\nabla_{\mathbf{z}_t}\text{Penalty}(\text{DEC}(\mathbf{z}_t), \mathcal{W})$, making the guidance possible. The full algorithm for Luminark applied to latent diffusion models is provided in Algorithm 4.

---

**Algorithm 4** Luminark Injection for LDMs

---

1: **Input**: Denoiser model $D(\mathbf{z}, \sigma)$, Decoder DEC, diffusion steps $T$, noise schedule $\{\sigma_t\}_{t=0}^{T}$, watermark $\mathcal{W} = (\mathbf{c}, \boldsymbol{\tau})$, watermark guidance scale $s$, threshold $T_{\text{match}}$
2: **repeat**
3:     **Initialize**: sample $\mathbf{z}_T \sim \mathcal{N}(0, \sigma_T^2 \mathbf{I})$
4:     **for** $t = T, \ldots, 1$ **do**

5:         $\mathbf{z}_{t-1} \leftarrow \mathbf{z}_t - \left[ \underbrace{\dfrac{\mathbf{z}_t - D(\mathbf{z}_t, \sigma_t)}{\sigma_t}}_{\text{Denoising Term}} + \underbrace{s\nabla_{\mathbf{z}_t}\text{Penalty}(\text{DEC}(\mathbf{z}_t), \mathcal{W})}_{\text{Watermark Guidance Term}} \right] (\sigma_t - \sigma_{t-1})$

6:     $\mathbf{x}_0 \leftarrow \text{DEC}(\mathbf{z}_0)$         ▷ Decode final latent to image
7:     $m \leftarrow \frac{1}{N}\sum_{i=1}^{N} \mathbb{I}[\text{sgn}(l(\mathbf{p}_i) - \tau_i) = c_i]$     ▷ Compute Match Rate on $\mathbf{x}_0$
8: **until** $m \geq T_{\text{match}}$
9: **return** $\mathbf{x}_0$

---

## E GUIDED SAMPLING FOR AR MODELS

Recent works, including the Vision Autoregressive Model (VAR) (Tian et al., 2024) and Masked Autoregressive Model (MAR) (Li et al., 2024), utilize guidance to enhance the quality of generation. For context, the autoregressive model can be generally formulated as:

$$P(\mathbf{h}_1, \mathbf{h}_2, \cdots, \mathbf{h}_T) = \prod_{k=1}^{T} P(\mathbf{h}_t | \mathbf{h}_{<t}), \tag{8}$$

where $\mathbf{h}_{<t}$ denotes $(\mathbf{h}_1, \mathbf{h}_2, \ldots, \mathbf{h}_{t-1})$. In VAR, $\mathbf{h}_t$ corresponds to the intermediate $t$-th-resolution images in the generation process, whereas in MAR, $\mathbf{h}_t$ is defined as the set of image tokens generated at the $t$-th step.

In the diffusion paradigm, Watermark Guidance is applied by modifying the denoising term at every step to enforce the luminance constraint, with its core idea being a step-wise modification procedure. Autoregressive models also proceed sequentially, which allows any guidance to be incorporated by adjusting the distribution $p(\mathbf{h}_t \mid \mathbf{h}_{<t})$ at each step. Since both VAR and MAR operate in the latent space, we adopt the same methodology described in Appendix D: at each step, the latent representation $\mathbf{z}$ is decoded into an image via DEC, then we can obtain $\nabla_{\mathbf{z}}\text{Penalty}(\text{DEC}(\mathbf{z}), \mathcal{W})$. Detailed procedures are outlined in Algorithms 5 and 6, where modifications to the original code are highlighted in blue.

---

**Algorithm 5** Luminark Injection for VAR

---

1: **Input**: Autoregressive model $p$, Decoder DEC, watermark pattern $\mathcal{W} = (\mathbf{c}, \boldsymbol{\tau})$, watermark guidance scale $s$, label $y$, resolution_scales (e.g., $\{1, 2, 3, 4, 5, 6, 8, 10, 13, 16\}$), apply_constraint_steps (e.g., $\{8, 10, 13, 16\}$), guidance steps $S$, threshold $T_{\text{match}}$
2: **repeat**
3:     **Initialize**: $\mathbf{z} \leftarrow \text{embedding}(y)$         ▷ VAR employs conditional generation by default
4:     **for** $t = 1, 2, \cdots, \text{len}(\text{resolution\_scales})$ **do**
5:          $\tilde{\mathbf{h}}_{t-1} \leftarrow \text{interpolate}(\mathbf{z}, \text{resolution\_scales}[t-1])$      ▷ Interpolate $\mathbf{z}$ to current resolution
6:          Sample next token $\mathbf{h}_t \sim p(\mathbf{h}_t | \tilde{\mathbf{h}}_0, \tilde{\mathbf{h}}_1, \cdots, \tilde{\mathbf{h}}_{t-1})$
7:          last_scale $\leftarrow$ resolution_scales$[-1]$      ▷ The final image scale (i.e., the scale of $\mathbf{z}$)
8:          $\mathbf{z} \leftarrow \mathbf{z} + \text{interpolate}(\mathbf{h}_t, \text{last\_scale})$         ▷ Residual design
9:          **if** t in apply_constraint_steps **then**
10:             $\mathbf{z} \leftarrow \mathbf{z} - \nabla_{\mathbf{z}} \text{Penalty}(\text{DEC}(\mathbf{z}), \mathcal{W})$      ▷ Apply guidance
11:      $\mathbf{x} \leftarrow \text{DEC}(\mathbf{z})$         ▷ Decode final latent to image
12:      $m \leftarrow \frac{1}{N} \sum_{i=1}^{N} \mathbb{I}[\text{sgn}(l(\mathbf{p}_i) - \tau_i) = c_i]$      ▷ Compute Match Rate on $\mathbf{x}$
13: **until** $m \geq T_{\text{match}}$
14: **return** $\mathbf{x}$

---

**Algorithm 6** Luminark Injection for MAR

---

1: **Input**: Autoregressive model $p$, Decoder DEC, watermark pattern $\mathcal{W} = (\mathbf{c}, \boldsymbol{\tau})$, watermark guidance scale $s$, label $y$, ar_step $T$, threshold $T_{\text{match}}$
2: **repeat**
3:     **Initialize**: mask_steps $\leftarrow$ gen_mask_order$(T)$      ▷ Random generation order
4:     **Initialize**: $\mathbf{z} \leftarrow \mathbf{0}$      ▷ Maintain the generated $\mathbf{h}$ with corresponding position
5:     **for** $t = 1, 2, \cdots, \text{ar\_step}$ **do**
6:          mask_to_pred $\leftarrow$ mask_steps[t]
7:          Sample next tokens $\mathbf{h}_t \sim p(\mathbf{h}_t | \mathbf{h}_1, \mathbf{h}_2, \cdots, \mathbf{h}_{t-1}, y, \text{mask\_to\_pred})$
8:          $\mathbf{z}[\text{mask\_to\_pred}] \leftarrow \mathbf{h}_t$      ▷ Update masked positions in $\mathbf{z}$
9:          $\mathbf{h}_t \leftarrow \mathbf{h}_t - \nabla_{\mathbf{h}_t} \text{Penalty}(\text{DEC}(\mathbf{z}), \mathcal{W})$      ▷ Apply guidance
10:          $\mathbf{z}[\text{mask\_to\_pred}] \leftarrow \mathbf{h}_t$      ▷ Update masked positions in $\mathbf{z}$
11:      $\mathbf{x} \leftarrow \text{DEC}(\mathbf{z})$      ▷ Decode final latent to image
12:      $m \leftarrow \frac{1}{N} \sum_{i=1}^{N} \mathbb{I}[\text{sgn}(l(\mathbf{p}_i) - \tau_i) = c_i]$      ▷ Compute Match Rate on $\mathbf{x}$
13: **until** $m \geq T_{\text{match}}$
14: **return** $\mathbf{x}$

---

## F   Description of Baseline Methods

To the best of our knowledge, few studies have investigated watermarking techniques on state-of-the-art autoregressive models, such as VAR (Tian et al., 2024) and MAR (Li et al., 2024). Notable watermarking approaches are specifically designed for diffusion models, such as recent Gaussian-Shading (Yang et al., 2024) and the PRC watermark (Gunn et al., 2024). Therefore, we select these two methods as the baseline.

Both Yang et al. (2024) and Gunn et al. (2024) are developed upon the pioneer Tree-Ring (Wen et al., 2023) approach. The core principle of these methods is to inject a signature into the initial noise during the diffusion process, and subsequently generate the watermarked image by solving the Probability-Flow Ordinary Differential Equation (ODE). The verification process leverages the diffeomorphism property of the ODE: we can use an inverse ODE to reconstruct the initial noise, thereby detecting the watermark's presence. Specifically, the signature in Gaussian Shading consists of noise samples drawn from specific intervals of the Gaussian distribution, while that of PRC-W is defined as the signs of the noise vector. From the description, we can also easily see that this mechanism is specifically designed within the diffusion paradigm and incompatible with other generative architectures.

Compared with Yang et al. (2024) and Gunn et al. (2024), a key difference in our experimental setup is that we generate a unique watermark for each method and keep it fixed throughout the experiment. In contrast, the experimental setup in Yang et al. (2024) and Gunn et al. (2024) randomly generates a watermark on the fly for each image. Our setting is evidently more practical, as a service provider can realistically maintain only a limited number of keys, which makes reliable detection feasible. By comparison, generating keys on the fly not only incurs significant overhead in maintaining keys but also complicates detection. A notable advantage of this alternative experimental design is that it increases image diversity, which in turn yields substantially lower FID scores. For completeness, we also replicated their experimental setting. Empirically, we observe that the FIDs of all methods are similar and close to the reference.

## G  HYPERPARAMETERS

### G.1  EDM2 HYPERPARAMETERS

We conduct experiments on three models from the EDM2 family: EDM2-XXL, EDM2-L, and EDM2-XS. Each model is a latent diffusion model trained on the ImageNet dataset, with images generated at a final resolution of $512 \times 512$.

All models in the EDM2 family utilize a U-Net architecture. For EDM2-XXL, the number of channels is set to 448, and the number of residual blocks per resolution in U-Net is set to 3, resulting in a total of 3.05 billion parameters. For EDM2-L, the number of channels is set to 320, and the number of blocks is set to 3, resulting in a total of 1.56 billion parameters. For EDM2-XS, the number of channels is set to 128, and the number of parameters is 0.25 billion. The diffusion process is performed in the latent space, whose dimensionality is $64 \times 64 \times 4$. Each image is first encoded into this space using a VAE encoder. After the diffusion process, the latent representation is projected back into the pixel space through the corresponding decoder.

We conduct our experiments using the official open-source implementation[3] and the released pre-trained checkpoints. For the diffusion process, we follow the default configuration: the number of diffusion steps is fixed at 32, with $\sigma_{min} = 0.002$, $\sigma_{max} = 80$, and $\rho = 7$, employing Heun's second-order ODE solver. For our watermark generator, the threshold $\tau$ is randomly sampled from a uniform distribution $U(0.4, 0.6)$ to avoid extreme values, and $c$ is randomly chosen from -1 and +1 with equal probability. $\tau$ and $c$ are generated and then fixed during each experiment. For watermark injection, the patch size is a key hyperparameter. Choosing a patch size that is too small overly constrains local flexibility, whereas an excessively large patch size degrades detection performance. In all experiments, we therefore fix the patch size to $64 \times 64$ for all models, and the total number of patches is 64. The watermark guidance scale ($s$) controls the strength of the injection during the denoising process. For EDM2-XXL, we set $s$ to be 0.00082. For EDM2-L, we set $s$ to be 0.0084. For EDM2-XS, we set $s$ to be 0.0084. To achieve a false positive rate ($fpr$) of 1%, we set match rate threshold $T_{match} = 0.61$ using Algorithm 1.

### G.2  VAR HYPERPARAMETERS

We conduct experiments on three models from the VAR family: VAR-d36, VAR-d30, and VAR-d16. Each model is a latent visual autoregressive model trained on the ImageNet dataset, generating images at a final resolution of $512 \times 512$ for VAR-d36, and $256 \times 256$ for VAR-d16 and VAR-d30.

All models in the VAR family directly leverage GPT-2-like transformer architecture (Radford et al., 2019). In VAR-d36, the embedding dimension is 2304, with 36 attention heads and 36 transformer layers, resulting in a total of 2.35 billion parameters. In VAR-d30, the embedding dimension is 1920, with 30 attention heads and 30 transformer layers, resulting in a total of 2.01 billion parameters. In VAR-d16, the embedding dimension is 1024, with 16 attention heads and 16 transformer layers, resulting in a total of 0.31 billion parameters. The generation process is performed in the latent space, whose dimensionality is $32 \times 32 \times 32$ for VAR-d36, and $16 \times 16 \times 32$ for VAR-d16 and VAR-d30. The resolution scales used are $\{1, 2, 3, 4, 6, 9, 13, 18, 24, 32\}$ for VAR-36, while $\{1, 2, 3, 4, 5, 6, 8, 10, 13, 16\}$ for VAR-16 and VAR-30. Each image is first encoded into the latent

---

[3]https://github.com/NVlabs/edm2

space using a VQVAE encoder. After the generation process, the latent representation is projected back into pixel space through the corresponding decoder.

We conduct our experiments using the official open-source checkpoint [4]. Since Tian et al. (2024) only released a demo rather than the full sampling code, we re-implemented their method for comparison. For our watermark generator, the threshold $\tau$ is randomly sampled from a uniform distribution $U(0.4, 0.6)$ to avoid extreme values, and $c$ is randomly chosen from -1 and +1 with equal probability. $\tau$ and $c$ are generated and then fixed during each experiment. For watermark injection, we fix the total number of patches to $64$ for all models, resulting in a patch size of $64 \times 64$ for VAR-36, while $32 \times 32$ for VAR-16 and VAR-30. Watermark injection is not applied at all scales. For VAR-16 and VAR-30, the watermark constraint is enforced at the last four resolutions (8, 10, 13, and 16), whereas for VAR-36, the constraint is applied only at the final resolution (32). The watermark guidance scale is fixed at $s = 0.05$ for VAR-16 and VAR-30. For VAR-36, since the guidance is applied only at the final step, we adopt a gradient-descent formulation with a reduced strength of $s = 0.015$, optimizing over 8 steps. To achieve a false positive rate ($fpr$) of 1%, we set match rate threshold $T_{match} = 0.625$ using Algorithm 1.

### G.3  MAR Hyperparameters

We conduct experiments on three models from the MAR family: MAR-B, MAR-L, and MAR-H. Each model is a latent autoregressive model trained on the ImageNet dataset, with images generated at a final resolution of $256 \times 256$.

All models in the MAR family utilize a Transformer (Vaswani et al., 2017) ViT (Dosovitskiy et al., 2020) architecture for token embedding and an additional MLP for the diffusion procedure. MAR-H, MAR-L, and MAR-B, respectively, have 40, 32, and 24 Transformer blocks with a width of 1280, 1024, and 768. The denoising MLP, respectively, has 12, 8, and 6 blocks and a width of 1536, 1280, and 1024. The generation process is performed in the latent space, whose dimensionality is $16 \times 16 \times 16$. Each image is first encoded into this space using a VAE encoder. After the generation process, the latent representation is projected back into pixel space through the corresponding decoder.

We conduct our experiments using the official open-source implementation [5] and the released pretrained checkpoints. For the generation process, we follow the default configuration: the number of autoregressive steps is fixed at 256, and the diffusion sampling step is set to 100. For our watermark generator, the threshold $\tau$ is randomly sampled from a uniform distribution $U(0.4, 0.6)$ to avoid extreme values, and $c$ is randomly chosen from -1 and +1 with equal probability. $\tau$ and $c$ are generated and then fixed during each experiment. For watermark injection, the total number of patches is fixed at 64, yielding a patch size of $32 \times 32$, and the guidance scale is set to $s = 0.015$ across all models. To achieve a false positive rate ($fpr$) of 1%, we set match rate threshold $T_{match} = 0.61$ using Algorithm 1.

## H  Use of Large Language Models (LLMs)

In preparing this manuscript, we used ChatGPT (OpenAI) solely as a writing assistant to improve the readability and fluency of the text. Specifically, the model was employed to polish the language and refine grammar without contributing to the conceptualization, methodology, analysis, or results of this work. All ideas, experiments, and conclusions are entirely the authors' own.

---

[4]https://huggingface.co/FoundationVision/var
[5]https://github.com/LTH14/mar

