# OpenReview forum: "Luminark: Training-free, Reliable Watermarking for General Vision Generative Models"
_ICLR.cc/2026/Conference — Submitted to ICLR 2026_

### Official Review · Reviewer_FRcA · 2025-10-23

**Soundness:** 3
**Presentation:** 3
**Contribution:** 4
**Rating:** 4
**Confidence:** 4

**Summary:**

The paper introduces Luminark, a training-free watermarking method for vision generative models that embeds imperceptible signatures via patch-level luminance statistics. A watermark is defined as a secret binary pattern determined by whether each image patch's average luminance exceeds predefined thresholds. Injection leverages guidance techniques as a plug-and-play mechanism to enforce this pattern during generation across diffusion, autoregressive, and hybrid models without degrading visual quality. Detection computes the match rate between the extracted binary pattern and the secret one, with theoretical bounds on false positives.

**Strengths:**

1.  Good Paper Writing: The paper is exceptionally well-written, featuring a clear structure and a detailed description of the methodology. Furthermore, the authors provide sound theoretical arguments to justify the statistical properties of their detection mechanism.

2.  Novel Watermark Definition Proposed: The paper introduces a novel and insightful watermarking strategy based on patch-level luminance statistics. This approach, which defines the signature based on patch-level mean luminance constraints, represents a new and creative direction for watermark design.

3.  General-Purpose and Training-Free Injection: The method features an elegant "plug-and-play" design. The authors insightfully leverage the guidance mechanism, a common component in modern generative models, to "softly" inject the watermark at runtime. This training-free approach successfully enables the method's broad effectiveness and cross-model generality.

**Weaknesses:**

1. Concerns Regarding Robustness: This study lacks experiments focused on advanced editing methods, such as robustness testing for image editing techniques based on diffusion models. The proposed method heavily relies on the stability of patch brightness averages. If I were to apply regional editing (e.g., composition or inpainting) to images with LUMINARK brightness watermarks, or perform specific global editing (e.g., regeneration or style transfer), I suspect this could significantly undermine the detectability of the method. However, I acknowledge that this is the first research of its kind, and inspiring the community is more important than achieving perfection. Thus, some limitations are acceptable;

2. Experiments about Baseline Comparison: My second concern relates to the baselines used for comparison. The evaluated baselines include DWTDCT, DWTDCTSVD, RivaGAN, and methods specifically designed for diffusion-specifics. However, some of the baseline watermarking techniques are outdated, particularly the post-processing methods. It remains uncertain whether the proposed method can effectively counter modern post-processing watermarks, such as TrustMark (ICCV 2025) and VINE (ICLR 2025). To ensure a more comprehensive evaluation, experiments involving at least VINE should be included to obtain more convincing results (p.s., I understand this may be a lot to run, but some small yet convincing experiments would really help).

**Questions:**

Regarding Robustness Against AI-Driven Edits: As mentioned in the Weakness, the paper's robustness evaluation focuses on traditional transformations. Could the authors provide experiments comparing Luminark against a state-of-the-art baseline like VINE (ICLR 2025) on advanced diffusion-based image editing cases? (VINE and its associated W-Bench are specifically designed to benchmark robustness against advanced, AI-driven image editing (e.g., regeneration, inpainting). A direct comparison with it would be crucial to substantially validate Luminark's effectiveness against this modern and highly relevant threat attacks.

Regarding the Hyperparameter Choice: How does the method handle variable image resolutions? Is there a dynamic way to set $k$? Also, have the authors considered using hierarchical or overlapping patches to improve robustness?

---

### Official Review · Reviewer_kwxc · 2025-10-27

**Soundness:** 2
**Presentation:** 2
**Contribution:** 2
**Rating:** 4
**Confidence:** 3

**Summary:**

The paper proposes Luminark, a training-free watermarking method for vision generative models that encodes a binary signature in patch-level luminance statistics and injects it via a plug-and-play guidance term during sampling. Detection compares the image’s patchwise luminance-threshold exceedances to a secret binary pattern and offers a simple statistical control of false positives. The method is evaluated on nine models spanning diffusion (EDM2), autoregressive (VAR), and hybrid (MAR), showing high detection accuracy under a broad set of image transformations and FID close to unwatermarked references.

**Strengths:**

1) Works across diffusion, AR, and hybrid models via a standard guidance hook; does not require retraining or model-specific reverse solvers.
2) The match-rate detector has a clear statistical guarantee and a practical calibration procedure for FPR.
3) Strong detection under common image edits and small FID degradation relative to unwatermarked generations.

**Weaknesses:**

1) The method may require repeated generations until the match rate exceeds the threshold and per-step backprop for the penalty, which can be expensive for high-resolution or long sampling schedules. The authors acknowledge this as a limitation. Quantifying overhead and proposing concrete speedups would strengthen the work. What are the typical additional compute costs (wall-clock, number of backward passes per step, average number of resamples) across models and resolutions? Please provide a cost-quality-detection curve.

2) The detector relies on a secret random pattern and thresholds. The paper would benefit from a threat-model discussion: key reuse across deployments, risk of adaptive adversaries who can query a detector, and the feasibility of targeted attacks (e.g., learning to flip just-enough patches without visible artifacts).

3) Synchronization issues under spatial transforms: Although the authors discuss robustness to many transforms and suggest OR-ing detections on flipped images, more systematic handling of spatial permutations (flips, rotations, translations, rescaling with different alignments) and patch-grid synchronization would be helpful. Are multi-view detectors or transform-invariant patterns necessary?

4) On text-to-image or class-conditional sampling, the guidance competes with semantic conditioning. Empirical results are positive, but analysis of failure modes (content drift, artifact risks at high guidance strength) would improve confidence.

5) The evaluated baselines include DWTDCT, DWTDCTSVD, RivaGAN, and methods specifically designed for watermarking generative images, such as GS PRC-W. However, the baselines for watermarking arbitrary images are outdated. Incorporating more recent methods, such as TrustMark [1] and VINE [2], which may perform better in testing, would make the comparison more thorough and robust.

6) Comparisons to training-based methods: While the paper motivates against fine-tuning approaches, a lightweight finetune baseline (e.g., few steps, small subset) could contextualize the train-free trade-offs.

7) The method effectively encodes a fixed-size binary pattern over patches. A discussion of capacity vs. image size, patch size, and desired FPR/TPR would clarify limits and scalability.

8) Have you experimented with mixed guidance (e.g., classifier-free + watermark) and schedules adapting the watermark scale over time?

[1] TrustMark: Universal Watermarking for Arbitrary Resolution Images

[2] Robust Watermarking Using Generative Priors Against Image Editing: From Benchmarking to Advances

**Questions:**

See weaknesses.

---

### Official Review · Reviewer_zoHK · 2025-10-29

**Soundness:** 3
**Presentation:** 3
**Contribution:** 2
**Rating:** 6
**Confidence:** 3

**Summary:**

The paper aims to solve the challenge of embedding reliable, robust, and general-purpose watermarks into images generated by vision generative models. Luminark is a training-free watermarking algorithm utilizes patch-level luminance statistics for watermark encoding. It also leverages widely adopted guidance techniques so the process is agnostic to network architectures. The experiments demonstrate that the watermark is imperceptible to humans, robust against common image transformations (compression, smoothing, quantization, noise, etc.), and can be reliably detected with statistical guarantees.

**Strengths:**

**Originality:**

The paper introduces a novel, training-free watermarking method leveraging patch-level luminance statistics, enabling universal application across generative model paradigms. This removes model-specific and fine-tuning constraints inherent in prior methods.

**Quality:**

The work is methodologically rigorous, providing statistical guarantees, comprehensive ablations, and benchmarking against established baselines across diverse state-of-the-art models. Experimental results robustly validate both robustness and detection accuracy.

**Clarity:**

The exposition is clear and technically precise, with formal definitions, intuitive illustrations, and clearly described algorithmic procedures that facilitate reproducibility.

**Significance:**

Luminark provides a robust, imperceptible, and general watermarking solution for vision generative models. Its paradigm-agnostic, plug-and-play design has substantial implications for digital content protection and AI safety.

**Weaknesses:**

- As suggested in the conclusion, Luminark’s injection process necessitates repeated image generation and additional backpropagation for guidance, which significantly raises computational costs compared to post-hoc watermarking or training-based approaches. This may hinder large-scale deployment. Future work should focus on more efficient penalty functions and optimization strategies.
- While the paper reports FID scores to quantify perceptual quality, these metrics have known limitations in capturing nuanced human judgments, especially for subtle artifacts introduced by watermarking. To strengthen the claim of imperceptibility and robust fidelity, it is recommended that the authors conduct human-based evaluation experiments (e.g., user studies, pairwise comparisons, or psychophysical tests). Such assessments would provide direct evidence of the watermark’s impact on user experience and complement the quantitative results for a more comprehensive validation.
- This work introduces the technically novel idea of applying patch-level luminance-based processing for watermark embedding across generative models. However, the underlying motivation for this approach remains somewhat unclear, particularly given the additional computational complexity it entails. While the method effectively unifies watermarking across multiple architectures, its real-world applicability appears constrained by these computational demands. Thus, although the paper offers a noteworthy knowledge contribution, its immediate practical impact seems limited.

**Questions:**

- The watermarking mechanism based on patch-level luminance statistics relies on adjusting the brightness of image patches to encode information. A potential failure case is when this process unintentionally leads to visible brightness artifacts or inconsistencies, thereby degrading the perceptual quality of the generated images. Do you think this risk exists? If so, how to prevent this from happening?
- The authors primarily evaluate their method on EDM2, VAR, and MAR diffusion models.  Why not choose something more popular like Stable Diffusion models.

---

### Official Review · Reviewer_tSu2 · 2025-10-31

**Soundness:** 2
**Presentation:** 3
**Contribution:** 2
**Rating:** 2
**Confidence:** 5

**Summary:**

This paper introduces Luminark, a training-free watermarking method for generative vision models. The watermark is defined as a predefined binary pattern based on patch-level luminance statistics. To embed the watermark, the method uses "watermark guidance", a mechanism that steers the generative process (in diffusion, autoregressive, or hybrid models) to produce an image whose patch-level luminance matches the target pattern. Detection is performed by partitioning a query image into patches, calculating the luminance, and checking if the resulting binary pattern matches the predefined one.

**Strengths:**

S1. Generality: The core idea of using a guidance-based mechanism is good, as it allows the method to be applied across different generative paradigms, including diffusion (EDM2), autoregressive (VAR), and hybrid (MAR) models, without model-specific modifications.

S2. Thorough Experiments: The method is evaluated on a wide range of nine models , covering different architectures (U-Nets, Transformers), resolutions (256x256 and 512x512), and model scales. The experiments are non-trivial and use state-of-the-art generative frameworks.

**Weaknesses:**

W1. Weak Baselines.
The paper does not provide evidence that Luminark is superior to strong, modern post-hoc watermarking methods. The baselines used (DwtDct, DwtDctSvd, RivaGAN) are outdated and known to be non-robust.
To ensure a fair comparison, stronger and more recent baselines such as TrustMark, WAM, and Video Seal could be included.

W2. Insufficient Robustness Evaluation.
The evaluation of robustness against geometric transformations is inadequate.
The paper’s “Cropping” attack only removes a 2-pixel border, which is not a realistic scenario.
Robustness should be evaluated against more significant geometric attacks such as: random crop-and-resize, rotation, perspective transformations, hflip, etc.

W3. Misrepresentation of Prior Work
The paper misunderstands several training-based watermarking methods in the Introduction and Related Work sections. Namely, (Fernandez et al., 2023; Min et al., 2024) do not work by fine-tuning model weights on pre-watermarked images.

W4. Suboptimal Statistical Analysis. The statistical analysis presented in Proposition 1 is weak.
The match rate $m(x, \mathcal{W})$ should follow a binomial distribution, for which an exact p-value or a tight confidence interval (e.g., Clopper–Pearson) can be computed. However, the paper instead applies Hoeffding’s inequality, which provides only a loose bound and suggests a misunderstanding of the underlying statistical test.

**Questions:**

Q1 (W1): Why were state-of-the-art post-hoc watermarking methods, which (like Luminark) are training-free and model-agnostic, omitted from the experimental comparison?

Q2 (W2): How does Luminark's detection accuracy perform under more realistic geometric attacks, such as random cropping (e.g., 50% area) followed by resizing, or rotations of more than a few degrees?

Q3 (W4): Given that the match rate follows a binomial distribution, why did the authors use an inequality bound (Proposition 1) and an empirical threshold (Algorithm 1)  instead of an exact binomial test, which would provide a reliable p-value without needing a large dataset of unwatermarked images to calibrate?

---

### Author Response · Authors · 2025-11-28
**General Response**

We sincerely thank all reviewers and Area Chair for your constructive comments and suggestions.

After reviewing the feedback, we believe we can address most of the concerns raised, however, it would require adding significant content to the manuscript, thus increasing the paper’s length. It is not feasible to reasonably condense the revised manuscript to meet the ICLR length constraints within the short rebuttal timeframe without compromising the paper’s clarity and coherence.

We will refine our manuscript according to your suggestions and prepare a more comprehensive version for a future conference.

---

### Meta-Review · Area_Chair_gxMd · 2026-01-05

**Summary:**

The paper introduces Luminark, a training-free watermarking method for vision generative models that encodes a secret binary signature using patch-level luminance statistics. The watermark is injected via a plug-and-play guidance term during generation, enabling use across diffusion, autoregressive, and hybrid models without modifying model weights or degrading visual quality. Experiments on nine state-of-the-art models demonstrate high detection accuracy, robustness to common image transformations, and controlled false positive rates. The authors have decided to submit to another conference.

**Reviewer Concerns:**

Reviewer tSu2: The main concerns are weak and outdated baseline comparisons, insufficient robustness evaluation against realistic geometric attacks, and an inappropriate statistical analysis that should use exact binomial tests rather than loose concentration bounds.

Reviewer zoHK: The reviewer is primarily concerned about the high computational overhead of guidance-based injection and the lack of human perceptual studies to convincingly support claims of imperceptibility and practical impact.

Reviewer kwxc: The key issues are unclear computational cost–quality trade-offs, an underdeveloped threat model (e.g., adaptive adversaries and key reuse), limited handling of spatial transformations, and missing comparisons to stronger modern baselines.

Reviewer FRcA: The main concern is the lack of evaluation against advanced AI-driven image editing attacks (e.g., diffusion-based inpainting or regeneration), along with insufficient comparison to recent robust post-hoc watermarking methods such as VINE or TrustMark.

**Reviewer Scores:**

The authors decided to submit to another conference so I do not think the scores would have changed.

---

### Decision · Program_Chairs · 2026-01-26

Reject